# ACHIEVING FAIRNESS-UTILITY TRADE-OFF THROUGH DECOUPLING DIRECT AND INDIRECT BIAS

## ABSTRACT

Fairness in regression is crucial in high-stakes domains such as healthcare, finance, and criminal justice, where biased predictions can perpetuate unequal treatment. Bias arises both directly, when sensitive attributes explicitly affect predictions, and indirectly, when correlated predictors act as proxies. Existing fairness-aware regression methods typically address only one type of bias or suffer from reduced predictive performance, especially in case of multivariate sensitive attributes. We introduce a fairness framework that adapts subspace decomposition techniques from envelope regression. The predictor space is decomposed into four orthogonal components: response-specific variation, sensitive variation, shared variation, and residual noise. By penalizing only the sensitive component, our approach offers interpretable control over the fairness-utility trade-off. Unlike black-box methods, it yields interpretable estimators with provable efficiency gains. We validate the framework through simulations and real-world experiments, demonstrating improved fairness and predictive accuracy compared to prior methods. Our results highlight predictor-space decomposition as a principled tool for building fair, efficient, and interpretable regression models.

## 1 INTRODUCTION

Machine learning is increasingly deployed in high-stakes domains such as healthcare, finance, and criminal justice (Das et al., 2021; Bogen & Rieke, 2018; Kourou et al., 2015; De Fauw et al., 2018; Raji & Buolamwini, 2019; Buolamwini & Gebru, 2018), where unfair predictions can reinforce or amplify social inequities. Ensuring fairness in predictive models is therefore essential. In regression, unfairness arises through two channels: (i) *direct bias*, when sensitive attributes (e.g., race, gender) directly affect predictions, and (ii) *indirect bias*, when correlated predictors act as proxies (Barocas et al., 2023; Calmon et al., 2017; Feldman et al., 2015). Most existing methods mitigate only one form of bias, rely on restrictive assumptions, or sacrifice predictive accuracy.

Naively removing sensitive variables fails to eliminate indirect bias and obscures how unfair influence enters the model. More sophisticated strategies - pre-processing, in-processing, and post-processing (Calders et al., 2013; Johnson et al., 2016; Komiyama et al., 2018; Berk et al., 2021; Agarwal et al., 2019)- have been developed, but primarily in the classification setting. Fair regression with continuous outcomes remains comparatively underexplored (Komiyama et al., 2018; Scutari et al., 2022), especially in the presence of multiple sensitive attributes and complex interactions. Existing regression-based approaches typically rely on constrained optimization or ad hoc penalization. These methods lack a principled decomposition of the predictor space to distinguish direct from indirect bias and do not exploit opportunities for statistical efficiency.

We address these gaps with the *Fair Envelope Regression Model* (FERM) framework. FERM leverages envelope regression to decompose the predictor space into four orthogonal components: (i) response-only, (ii) sensitive-only, (iii) shared response–sensitive, and (iv) residual. Penalizing only the sensitive components provides interpretable control over the fairness–utility trade-off, allowing practitioners to impose fairness constraints without discarding predictive signal. The envelope structure further improves estimation efficiency, yielding more stable estimates than existing fairness-aware baselines. We provide theoretical guarantees on consistency and efficiency, provide expressions detailing how fairness-utility trade-off is achieved, and validate FERM through simu-

lations and real-world data. Across settings, FERM consistently achieves superior fairness-utility trade-offs compared to prior methods.

OUR CONTRIBUTIONS

1. **Subspace Decomposition:** We propose FERM, a fairness-aware envelope regression framework that decomposes predictors into response-only, sensitive-only, shared, and residual components, enabling transparent attribution of sensitive influence.

2. **Fairness-Utility Trade-off:** By applying a ridge penalty only to sensitive subspaces, FERM provides a tunable mechanism that interpolates between unconstrained accuracy and full fairness.

3. **Efficiency and Theory:** FERM leverages the envelope structure to reduce asymptotic variance relative to the ordinary least squares (OLS) estimates, yielding more stable estimates, with formal guarantees of consistency and efficiency. We also provide a closed-form characterization of the fairness–utility trade-off.

4. **Empirical Validation:** Simulations and real-world experiments show that FERM achieves improved fairness-utility trade-offs compared to prior regression-based methods.

The remainder of this paper is organized as follows. Section 2 reviews related work on fairness in regression. Section 3 provides background on envelope regression methodology and related fairness approaches. Section 4 introduces the FERM framework, with theoretical results presented in Section 5. Section 6 evaluates the proposed method through a series of simulations and real-world datasets, benchmarking FERM against existing fairness-aware regression techniques such as Fair Ridge Regression Model (FRRM) (Scutari et al., 2022). Finally, Section 7 discusses limitations and outlines directions for future research.

## 2 RELATED WORK

Developing predictors that exhibit independence from protected attributes, often formalized through notions such as statistical parity, has been a central theme in algorithmic fairness research. Existing approaches can broadly be categorized into *in-processing*, *pre-processing*, and *post-processing* strategies, with the majority of work focusing on classification rather than regression. Below we review the most relevant directions for fairness in regression.

**Moment-based and linear regression approaches.** Early works such as Calders et al. (2013), Johnson et al. (2016), and Komiyama et al. (2018) enforce restricted independence through moment constraints, typically ensuring that predictions are uncorrelated with sensitive attributes. These methods are designed primarily for least squares regression and can handle both continuous and categorical attributes. Komiyama et al. (2018) in particular introduced a quadratic optimization framework that bounds the relative proportion of variance explained by sensitive attributes, offering explicit user control of fairness levels and theoretical optimality guarantees.

**In-processing with fairness constraints.** Fairness penalties have also been embedded directly into the regression objective. Berk et al. (2021) proposed convex formulations incorporating both individual and group fairness notions. Similarly, Pérez-Suay et al. (2017) enforced zero correlation in reproducing kernel Hilbert spaces (RKHS), though their method is largely restricted to least squares settings. More recently, Scutari et al. (2022) introduced a fairness-aware regression model with ridge penalties on sensitive attributes, yielding mathematically simple formulations, partially closed-form solutions, and extensions to generalized and kernelized regression.

**Post-processing and minimax analyses.** Post-processing strategies have been explored to enforce fairness after model training. For example, Chzhen et al. (2020) used Wasserstein barycenters, while Zhao (2021) derived tight lower bounds on the fairness–accuracy trade-off. Du et al. (2022) accounted for sample selection bias, and Taturyan et al. (2024) and Divol & Gaucher (2024) developed post-processing and unawareness-based methods that achieve demographic parity without requiring sensitive attributes at inference, an appealing property for privacy-constrained settings. Minimax analyses have further characterized optimal risks under fairness constraints (Chzhen &

Schreuder, 2022; Fukuchi & Sakuma, 2023), highlighting fundamental trade-offs as a function of feature dimension and group counts.

**Kernel and probabilistic approaches.** Several works extend fairness constraints to nonlinear settings. Kernel-based methods for equalized odds and mean-parity were introduced by Perez-Suay et al. (2023) and Wei et al. (2023), providing closed-form solutions in RKHS. Probabilistic models enforcing statistical independence were explored by Kamishima et al. (2012) and Fukuchi et al. (2015), though these often suffer from computational inefficiency and lack statistical guarantees. Finally, Agarwal et al. (2019) generalized the reductions-based minimax optimization framework of Agarwal et al. (2018) to regression, offering flexible in-processing methods with fairness constraints.

Despite this broad literature, most fairness-aware regression methods lack a principled mechanism to disentangle *direct* and *indirect* effects of sensitive attributes. Existing approaches typically rely on constrained optimization or penalization, but do not exploit subspace decompositions that could yield both interpretability and efficiency gains. Our work addresses this gap by introducing a fairness-aware envelope regression framework that provides a transparent decomposition of predictor space, a tunable fairness-utility trade-off, and improved estimation precision.

## 3 Preliminaries

We begin by introducing the key notation used throughout the paper. Let the random tuple $(X, S, Y)$ belong to the space $\mathbb{R}^{d_X} \times \mathcal{S} \times \mathbb{R}$, where $X \in \mathbb{R}^{d_X}$ denotes the non-sensitive feature, $Y \in \mathbb{R}$ is the response, and $S \in \mathcal{S} \subseteq \mathbb{R}^{d_S}$ represents the sensitive attribute, which can be scalar or vector-valued. Let $n$ denote the number of observations in the dataset $\{(X_i, S_i, Y_i)\}_{i=1}^n$. A fairness-aware algorithm aims to provide an estimator $\hat{Y}(X, S)$ for $Y$, based on the input $(X, S)$, while satisfying predefined fairness criteria. For convenience, we define the following matrices, representing the $n$ samples stacked by rows: $\mathbf{X} = [X_1 \cdots X_n]^T \in \mathbb{R}^{n \times d_X}$ for non-sensitive feature, $\mathbf{Y} = [Y_1 \cdots Y_n]^T \in \mathbb{R}^{n \times 1}$ for the response variable, and $\mathbf{S} = [S_1 \cdots S_n]^T \in \mathbb{R}^{n \times d_S}$ for the sensitive attributes. We throughout assume that $S, X$ and $Y$ are centered (zero mean).

### 3.1 Fairness Criteria in Regression with Multivariate Sensitive Attributes

Fairness in regression is typically enforced by requiring statistical independence between predictions $\hat{Y}$ and sensitive attributes $S$. Two common operationalizations are: (1) *Uncorrelatedness*: $\text{Cov}(\hat{Y}, S) = 0$, ensuring linear independence; (2) *Bounded explanatory power*: limiting the variance in $\hat{Y}$ explained by $S$, often via an $R^2$ (cf. Eq. (2)) measure (Komiyama et al., 2018; Scutari et al., 2022). We adopt the $R^2$ criterion, which is especially well-suited for multivariate $S$. It aggregates their joint contribution into a single interpretable quantity, avoiding multiple pairwise constraints and providing a direct knob for tuning fairness. For completeness, Appendix C.3 shows how our framework can incorporate alternative notions, such as equality of opportunity, by redefining the fairness subspace and penalty. Du et al. (2022) review regression fairness notions and confirm that $R^2$ (with partial correlations as an alternative) is among the most widely adopted.

### 3.2 Previous Works

Existing approaches to fairness in regression models often aim to reduce the association between $\mathbf{X}$ and $\mathbf{S}$ by introducing auxiliary de-correlation steps. Notably, Komiyama et al. (2018) proposed a multivariate linear regression to model the relationship between predictors and sensitive attributes, and use residuals as decorrelated predictors in the subsequent step as follows:

$$\mathbf{X} = \mathbf{S}B + \mathbf{U}, \tag{1}$$

The ordinary least squares (OLS) solution and the residuals are computed as

$$\hat{\mathbf{B}}_{\text{OLS}} = (\mathbf{S}^T \mathbf{S})^{-1} \mathbf{S}^T \mathbf{X} \in \mathbb{R}^{d_S \times d_X}, \qquad \hat{\mathbf{U}} = \mathbf{X} - \mathbf{S}\hat{\mathbf{B}}_{\text{OLS}} \in \mathbb{R}^{n \times d_X}.$$

By construction, the residuals $\hat{U} \in \mathbb{R}^{d_X}$, rows of $\hat{\mathbf{U}}$, are orthogonal to $S$, satisfying $\text{Cov}(S, \hat{U}) = 0$. Using these residuals, Komiyama et al. (2018) define a regression model: $Y = \boldsymbol{\alpha}^\top S + \boldsymbol{\beta}^\top \hat{U} + \epsilon$,

where $\boldsymbol{\alpha} \in \mathbb{R}^{d_S}$ and $\boldsymbol{\beta} \in \mathbb{R}^{d_X}$ are coefficients associated with $S$ and $\hat{U}$, respectively. To ensure fairness, they constrain the variance explained by $S$ using an $R^2$-based measure:

$$R^2(\boldsymbol{\alpha}, \boldsymbol{\beta}) = \frac{\mathrm{Var}(\boldsymbol{\alpha}^\top S)}{\mathrm{Var}(\hat{Y})}, \tag{2}$$

where $\hat{Y}$ denotes the predicted outcome. A fairness parameter $r \in [0, 1]$ is used to bound $R^2(\boldsymbol{\alpha}, \boldsymbol{\beta})$, controlling the trade-off between fairness and predictive performance. The resulting optimization problem is:

$$\min_{\boldsymbol{\alpha}, \boldsymbol{\beta}} \mathbf{E}[(Y - \hat{Y})^2] \quad \text{subject to} \quad R^2(\boldsymbol{\alpha}, \boldsymbol{\beta}) \leq r.$$

This approach explicitly balances prediction accuracy and fairness by controlling the influence of sensitive attributes on the outcome. Building on this framework, Scutari et al. (2022) highlighted limitations in the nonconvex formulation and proposed an alternative constrained optimization approach termed *Fair Ridge Regression Model (FRRM)*. Their method penalizes the sensitive attribute coefficients ($\boldsymbol{\alpha}$) with a ridge penalty while leaving the other coefficients ($\boldsymbol{\beta}$) unconstrained. Specifically, they solve the following problem:

$$\min_{\boldsymbol{\alpha}, \boldsymbol{\beta}} \|\mathbf{Y} - \mathbf{S}\boldsymbol{\alpha} - \hat{\mathbf{U}}\boldsymbol{\beta}\|_2^2 + \lambda(r)\|\boldsymbol{\alpha}\|_2^2,$$

where $\lambda(r) \geq 0$ is the ridge penalty ensuring that $R^2(\boldsymbol{\alpha}, \boldsymbol{\beta}) \leq r$. By imposing a direct penalty on $\boldsymbol{\alpha}$, the FRRM simplifies the optimization process and ensures that fairness constraints are met while maintaining flexibility.

**Limitations** Scutari et al. (2022) highlight key limitations in Komiyama et al. (2018)'s approach, including its reliance on a nonconvex optimization problem that is computationally challenging in high-dimensional settings and its restriction to linear regression models. Additionally, the fairness constraint becomes undefined as $r \to 0$, causing numerical instability, and the coupling of coefficients $\alpha$ and $\beta$ complicates interpretation. To address these issues, Scutari et al. (2022) propose the Fair Ridge Regression Model (FRRM), extending fairness constraints to generalized linear models and kernel regression. However, challenges remain: (1) **Inefficiency in Auxiliary Models:** The decomposition of $X$ into components explained by $S$ and residuals relies on multivariate linear regression, ignoring correlations among predictors. This simplification reduces statistical efficiency and prediction accuracy. (cf. our Theorem 5.1) (2) **Loss of Interpretability:** Residual-based decomposition obscures the relationship between $X$ and $S$, making it harder to understand how sensitive attributes influence outcomes.

### 3.3 Towards a Principled Decomposition

These limitations motivate a framework that (i) provides a principled subspace decomposition of predictors, (ii) improves statistical efficiency, and (iii) preserves interpretability. We seek to explicitly characterize how $X$ relates to $Y$ and $S$. As illustrated in Figure 1, we envision $X$ as partitioned into four interpretable components: variation predictive of $Y$, variation associated with $S$, shared variation, and residual noise. Such a decomposition clarifies the pathways through which sensitive attributes affect predictions and establishes a natural foundation for models that achieve transparent and tunable fairness–utility trade-offs.

## 4 Methodology

Our approach balances fairness and predictive performance by decomposing the predictor space $X$ relative to both the response $Y$ and the sensitive attributes $S$. As illustrated in Figure 1, $X$ can be partitioned into four interpretable parts: shared with both $Y$ and $S$, unique to $Y$, unique to $S$, and residual.

**Projection structure:** Let $(\Gamma, \Gamma_0)$ and $(\Phi, \Phi_0)$ be orthogonal bases in $\mathbb{R}^{d_X}$, where $\Gamma$ spans directions of $X$ associated with $S$ and $\Gamma_0$ its invariant complement; $\Phi$ spans predictive directions for $Y$

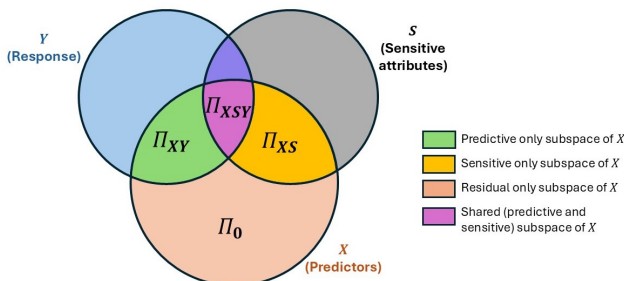

Figure 1: Conceptual decomposition of the predictor space $X$ relative to response $Y$ and sensitive attributes $S$. The diagram illustrates the partitioning of $X$-space into orthogonal subspaces: $\Pi_{XY}$ (predictive only), $\Pi_{XS}$ (sensitive only), $\Pi_0$ (residual variation in $X$) and $\Pi_{XSY}$ (shared predictive and sensitive).

and $\Phi_0$ the immaterial ones. The decomposition assumes

$$\Gamma_0^\top X \mid S \sim \Gamma_0^\top X, \quad \Gamma^\top X \perp\!\!\!\perp \Gamma_0^\top X \mid S, \qquad \text{with } \Gamma^\top \Gamma_0 = 0. \tag{3}$$

$$Y \perp\!\!\!\perp \Phi_0^\top X \mid \Phi^\top X, \quad \Phi^\top X \perp\!\!\!\perp \Phi_0^\top X, \qquad \text{with } \Phi^\top \Phi_0 = 0. \tag{4}$$

Intersecting these bases yields four orthogonal subspaces:

$$\begin{aligned}
\Pi_{XSY} &= \operatorname{span}(\Gamma) \cap \operatorname{span}(\Phi) && \text{(shared: predictive and sensitive)}, \\
\Pi_{XY} &= \operatorname{span}(\Gamma_0) \cap \operatorname{span}(\Phi) && \text{(predictive-only)}, \\
\Pi_{XS} &= \operatorname{span}(\Gamma) \cap \operatorname{span}(\Phi_0) && \text{(sensitive-only)}, \\
\Pi_0 &= \operatorname{span}(\Gamma_0) \cap \operatorname{span}(\Phi_0) && \text{(residual)}.
\end{aligned}$$

so any predictor decomposes as $X = X_{XSY} + X_{XY} + X_{XS} + X_0$, with $X_\bullet = P_{\Pi_\bullet} X$, where $P_{\Pi_\bullet}$ denotes the projection onto $\Pi_\bullet$.

**Regression models.** Using these components, we define:

$$\text{Fair model: } Y = \beta_{XY}^\top X_{XY} + \varepsilon, \tag{5}$$

$$\text{Unconstrained OLS: } Y = \beta_{XY}^\top X_{XY} + \beta_{XSY}^\top X_{XSY} + \varepsilon, \tag{6}$$

$$\text{Interpolated: } \hat{\beta} = \arg \min_{\beta_{XY}, \beta_{XSY}} \left\| \mathbf{Y} - \mathbf{X}_{XY} \beta_{XY} - \mathbf{X}_{XSY} \beta_{XSY} \right\|_2^2 + \lambda \| \beta_{XSY} \|_2^2. \tag{7}$$

Here $\lambda \geq 0$ tunes the fairness-utility balance: $\lambda \to \infty$ yields the fair model (5), $\lambda \to 0$ recovers OLS (6). We refer to these models as Fair Envelope Regression Models (FERM). Note that, FERM provides an explicit, auditable decomposition of the predictor space into four orthogonal components. Predictions use only the predictive subspace, with a controlled contribution from the shared subspace to achieve the desired fairness level. This makes it transparent which directions in $X$ drive prediction, which encode sensitive information, and how the fairness–utility trade-off is implemented - directly addressing Limitation (2) highlighted at the end of Section 3.2.

### 4.1 ALGORITHMIC IMPLEMENTATION

Envelope regression provides a practical way to estimate the bases $(\Gamma, \Gamma_0, \Phi, \Phi_0)$. Fitting a response envelope of $X$ relative to $S$ identifies directions associated with or invariant to $S$, while fitting a predictor envelope of $Y$ relative to $X$ isolates material and immaterial directions. Intersections of these envelopes yield empirical versions of $\Pi_{XSY}, \Pi_{XY}, \Pi_{XS}, \Pi_0$, enabling projection of $X$ into orthogonal components that disentangle bias and signal.

Algorithms 1 and 2 summarize the procedure: (i) estimate envelope bases and intersections; (ii) project $X$ into components; (iii) fit regression models with fairness–utility control by penalizing only the shared subspace. Technical details of envelope estimation, dimension selection, and ridge optimization are deferred to Appendix F.

---

**Algorithm 1** Envelope-Based Decomposition of the Predictor Space

---

1: **Input:** Training observations $\{(X_i, S_i, Y_i)\}_{i=1}^n$, with centered $X, S, Y$.
2: **Output:** Projection operators $\hat{\Pi}_{XSY}, \hat{\Pi}_{XS}, \hat{\Pi}_{XY}, \hat{\Pi}_0$.
3: **Response envelope for $X$ relative to $S$:** Estimate $\hat{\Gamma} \in \mathbb{R}^{d_X \times \hat{m}}$ and $\hat{\Gamma}_0$ (with $\hat{\Gamma}^\top \hat{\Gamma}_0 = 0$) by minimizing a standard response-envelope objective; select $\hat{m}$ by BIC or cross-validation.
4: **Predictor envelope for $Y$ relative to $X$:** Estimate $\hat{\Phi} \in \mathbb{R}^{d_X \times \hat{u}}$ and $\hat{\Phi}_0$ (with $\hat{\Phi}^\top \hat{\Phi}_0 = 0$) using a predictor-envelope objective; select $\hat{u}$ by BIC or cross-validation.
5: **Intersections (four subspaces):**
$$\hat{\Pi}_{XSY} := \mathrm{span}(\hat{\Gamma}) \cap \mathrm{span}(\hat{\Phi}), \quad \hat{\Pi}_{XY} := \mathrm{span}(\hat{\Gamma}_0) \cap \mathrm{span}(\hat{\Phi}),$$
$$\hat{\Pi}_{XS} := \mathrm{span}(\hat{\Gamma}) \cap \mathrm{span}(\hat{\Phi}_0), \quad \hat{\Pi}_0 := \mathrm{span}(\hat{\Gamma}_0) \cap \mathrm{span}(\hat{\Phi}_0).$$
6: **return** $\hat{\Pi}_{XSY}, \hat{\Pi}_{XS}, \hat{\Pi}_{XY}, \hat{\Pi}_0$.

---

**Algorithm 2** Training the Interpolated Regressor with Fairness Control

---

1: **Input:** Training observations $\{(X_i, S_i, Y_i)\}_{i=1}^n$, with centered $X, S, Y$; $\hat{\Pi}_{XSY}, \hat{\Pi}_{XS}, \hat{\Pi}_{XY}, \hat{\Pi}_0$ from Alg. 1;  target fairness $r \in [0, 1]$; new test point $(X_{new}, S_{new})$.
2: **Output:** Coefficients $(\hat{\beta}_{XY}, \hat{\beta}_{XSY})$, fitted predictor $\hat{Y}_\lambda$.
3: If $r = 0$, fit Fair model (5): $\hat{\beta}_{XY}^{\mathrm{fair}} := \arg\min_{\beta_{XY}} \|\mathbf{Y} - \mathbf{X}\hat{\Pi}_{XY}\beta_{XY}\|_2^2$;  $\hat{Y}_{\mathrm{fair}} := X_{XY}\hat{\beta}_{XY}^{\mathrm{fair}}$.
4: If $r = 1$, fit Unconstrained model (OLS) (6):
$$(\hat{\beta}_{XY}^{\mathrm{ols}}, \hat{\beta}_{XSY}^{\mathrm{ols}}) := \arg\min_{\beta_{XY}, \beta_{XSY}} \|\mathbf{Y} - \mathbf{X}\hat{\Pi}_{XY}\beta_{XY} - \mathbf{X}\hat{\Pi}_{XYS}\beta_{XSY}\|_2^2,$$
Compute $\hat{Y}_{new,\mathrm{OLS}} := (\hat{\beta}_{XY}^{\mathrm{OLS}})^\top \hat{\Pi}_{XY}^\top X_{new} + (\hat{\beta}_{XSY}^{\mathrm{OLS}})^\top \hat{\Pi}_{XSY}^\top X_{new}$.
5: If $r \in (0, 1)$, fit Interpolated (ridge) fit on shared component (7):
$$(\hat{\beta}_{XY}(\lambda), \hat{\beta}_{XSY}(\lambda)) = \arg\min_{\beta_{XY}, \beta_{XSY}} \left\|\mathbf{Y} - \mathbf{X}\hat{\Pi}_{XY}\beta_{XY} - \mathbf{X}\hat{\Pi}_{XYS}\beta_{XSY}\right\|_2^2 + \lambda\|\beta_{XSY}\|_2^2.$$
Here $\lambda$ is chosen such that $R_S^2(\lambda) \le r$ where
$$R_S^2(\lambda) := \frac{\mathrm{Var}(\hat{\beta}_{XSY}^\top(\lambda)X_{XSY})}{\mathrm{Var}(\hat{\beta}_{XSY}^\top(\lambda)X_{XSY} + \hat{\beta}_{XY}^\top(\lambda)X_{XY})}. \tag{8}$$
Compute $\hat{Y}_{new,\lambda} := \hat{\beta}_{XY}^\top(\lambda)\hat{\Pi}_{XY}^\top X_{new} + \hat{\beta}^\top(\lambda)_{XSY}\hat{\Pi}_{XSY}^\top X_{new}$.
6: **return** $(\hat{\beta}_{XY}(\lambda), \hat{\beta}_{XSY}(\lambda), \hat{Y}_\lambda)$.

---

## 5 THEORETICAL PROPERTIES

We now establish the key properties of our framework: (i) efficiency gains from modeling the predictor subspace structure, and (ii) closed-form characterizations of the fairness–utility trade-off. Prior residual-based approaches regress $Y$ on $X - X_{XS} = X_{XSY} + X_{XY} + X_0$, removing the sensitive-only component but still fitting OLS on directions that mix predictive, predictive–sensitive, and immaterial noise. In contrast, our method *exploits the envelope structure* by isolating the material subspace and regressing only on $X_{XSY} + X_{XY}$, thereby removing both sensitive-only and $Y$-immaterial components (Algorithm 2). This corresponds to projecting $\hat{\beta}_{\mathrm{OLS}}$ onto the predictive subspace $\mathrm{span}\{\Pi_{XSY}, \Pi_{XY}\}$, i.e. $\hat{\beta}_{\mathrm{env}} := P\hat{\beta}_{\mathrm{OLS}}$ is the estimated regression coefficient vector, where $P$ is the projection operator of the space spanned by $\Pi_{XSY}$ and $\Pi_{XY}$.

Let the asymptotic variance matrix, $avar(\cdot)$, such that if $\sqrt{n}(T - \theta) \to N(0, A)$, then $avar(\sqrt{n}T) = A$.

**Theorem 5.1** (Variance reduction via predictive projection). *Suppose* $\sqrt{n}(\hat{\beta}_{\mathrm{OLS}} - \beta) \to N(0, avar(\sqrt{n}\,\hat{\beta}_{\mathrm{OLS}}))$. *Then asymptotic covariance matrices satisfy*
$$avar(\sqrt{n}\,\hat{\beta}_{\mathrm{env}}) = P\,avar(\sqrt{n}\,\hat{\beta}_{\mathrm{OLS}})P \le avar(\sqrt{n}\,\hat{\beta}_{\mathrm{OLS}}).$$

*The inequality is strict whenever the residual subspace $\Pi_0$ is nontrivial.*

Theorem 5.1 shows that removing immaterial variation yields strictly lower asymptotic variance than OLS. This efficiency gain translates into improved predictive accuracy, as confirmed by our experiments in Section 6, directly addressing Limitation (1) highlighted at the end of Section 3.2.

**Proposition 5.2** (Consistency). *Suppose $\hat{\Sigma}_Y$, $\hat{\Sigma}_X$, $\hat{\Sigma}_{XY}$, $\hat{\Sigma}_{XS}$ are $\sqrt{n}$-consistent estimators for $\Sigma_Y, \Sigma_X, \Sigma_{XY}, \Sigma_{XS}$. Let $\hat{P}$ be the estimated projector obtained from Algorithm 1, and define $\hat{\beta}_{\mathrm{env}}(\hat{P}) = \hat{P}\hat{\beta}_{\mathrm{OLS}}$. Then $\hat{\beta}_{\mathrm{env}}$ is $\sqrt{n}$-consistent estimator for $\beta$.*

By construction, $\Pi_{XY}X$ is predictive yet independent of $S$. Furthermore, this implies that $\mathrm{Cov}(\hat{\Pi}_{XY}^\top X, S) \to 0$ as $n \to \infty$. Thus, if a linear regressor $\hat{Y}$ is constructed solely from $\hat{\Pi}_{XY}^\top X$, it achieves asymptotic fairness in the sense that $\mathrm{Cov}(\hat{Y}, S) \to 0$ as $n \to \infty$. We summarize this property in the following lemma.

**Lemma 5.3** (Asymptotically Fair Regressor). *Let $\hat{Y}_{\mathrm{fair}} := (\hat{\beta}_{XY}^{\mathrm{fair}})^\top X_{XY}$ with $\hat{\beta}_{XY}^{\mathrm{fair}}$ from (5). Under the assumptions of predictor-space decomposition, $\Pi_{XY}X \perp\!\!\!\perp S$ in the population, hence*

$$\mathrm{Cov}(\hat{Y}_{\mathrm{fair}}, S) \xrightarrow{p} 0 \quad as\ n \to \infty.$$

Finally, we characterize the interpolation between OLS and fair predictors under ridge penalization.

**Theorem 5.4** (Fairness-utility trade-off under predictor-space decomposition). *Assume that columns of $\mathbf{X}_{XSY}$ are orthonormal. Consider three fitted predictors: (i) $\hat{Y}_{fair}$ as in Lemma 5.3 ; (ii) $\hat{Y}_{OLS} = \hat{\beta}_{XY}^\top X_{XY} + \hat{\beta}_{XSY}^\top X_{XSY}$ after fitting (6); and (iii) the ridge predictor $\hat{Y}_\lambda = \hat{\beta}_{XY}(\lambda)^\top X_{XY} + \hat{\beta}_{XSY}(\lambda)X_{XSY}$ after fitting (7) with $\lambda = \lambda(r)$ for given unfairness level $r \in [0, 1]$ so that $R_S^2(\lambda) \le r$ where $R_S^2(\lambda)$ as in (8). Then the ridge predictor admits the closed-form representation*

$$\hat{Y}_\lambda = \frac{n}{n + \lambda(r)}\hat{Y}_{OLS} + \left(1 - \frac{n}{n + \lambda(r)}\right)\hat{Y}_{fair}, \qquad \lambda(r) \ge 0, \tag{9}$$

*interpolating between $\hat{Y}_{OLS}$ ($\lambda \to 0$) and $\hat{Y}_{fair}$ ($\lambda \to \infty$). Moreover, under squared loss its prediction risk decomposes as*

$$\mathbb{E}[(Y - \hat{Y}_\lambda)^2] = \mathbb{E}[(Y - \hat{Y}_{OLS})^2] + \left(1 - \frac{n}{n + \lambda}\right)^2 \mathbb{E}[(\hat{Y}_{OLS} - \hat{Y}_{fair})^2]. \tag{10}$$

This result quantifies the fairness-utility trade-off: penalizing the shared subspace smoothly interpolates between unbiased accuracy and fair model.

# 6 SIMULATION STUDY

We evaluate FERM against the FRRM model of Scutari et al. (2022). To demonstrate the value of decomposition, we present two variants of our method, corresponding to different subspace choices: (i) **FERM-decorrelated**: prediction is based on the de-correlated component of $X$ orthogonal to $S$ (i.e., variation in $X_{XS}$). This corresponds to removing the $S$-linked subspace before prediction. (ii) **FERM-predictive**: prediction is based on the subspace of $X$ relevant for $Y$ but orthogonal to $S$ (i.e., variation in $X_{XY}$). This leverages the predictor envelope to extract the most informative yet fair directions. In both cases, FERM enforces fairness by penalizing the sensitive subspace using a ridge penalty, with the penalty parameter chosen to satisfy a target unfairness budget.

The response variable is generated from the following model: $Y = \boldsymbol{\alpha}^\top S + \boldsymbol{\beta}^\top X + \epsilon$, where $\epsilon \sim \mathcal{N}(0, 0.5^2)$, $\boldsymbol{\alpha} \in \mathbb{R}^{d_S}$, and $\boldsymbol{\beta} \in \mathbb{R}^{d_X}$. The sensitive attributes $S$ are sampled independently from a multivariate standard normal distribution $\mathcal{N}(0_{d_S}, I_{d_S})$. The predictor variables $X$ consist of two components: one part correlated with $S$ and another part independent of $S$. Specifically, the correlated part of $X$, denoted as $X_{\mathrm{corr}} \in \mathbb{R}^{d_{\mathrm{corr}}}$, is generated by: $X_{\mathrm{corr}} = \beta_{SX}^\top S_Q S + \eta$, where $S_Q \in \mathbb{R}^{d_S \times d_S}$ is a projection matrix of rank $d_{XS}$ that captures the subspace of $X$ correlated with $S$, $\beta_{SX} \in \mathbb{R}^{d_S \times d_{\mathrm{corr}}}$ is the coefficient matrix, and $\eta \sim \mathcal{N}(0_{d_{\mathrm{corr}}}, I_{d_{\mathrm{corr}}})$ is an independent noise term. The independent part of $X$, denoted as $X_{\mathrm{indep}} \in \mathbb{R}^{d_X - d_{\mathrm{corr}}}$, is sampled independently from a multivariate standard normal distribution. The complete predictor matrix is constructed by concatenating

Table 1: Summary of simulation settings

| Setting | $d_X$ | $d_{corr}$ | $d_{XS}$ | $d_S$ | Sample size $n$ | Noise $\eta$ |
|---------|-------|-----------|----------|-------|-----------------|--------------|
| (1) | 40 | 20 | 5 | 10 | 5000 | normal |
| (2) | 100 | 50 | 15 | 20 | 20000 | normal |
| (3) | 40 | 20 | 5 | 10 | 5000 | poisson |

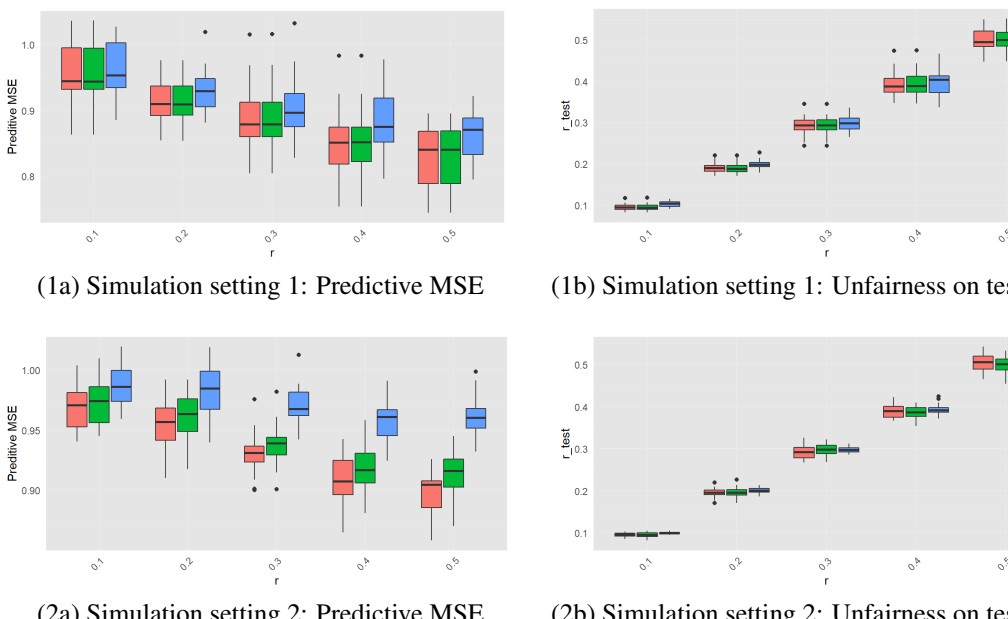

(1a) Simulation setting 1: Predictive MSE    (1b) Simulation setting 1: Unfairness on test data

(2a) Simulation setting 2: Predictive MSE    (2b) Simulation setting 2: Unfairness on test data

Figure 2: Left panel: Predictive MSE for FRRM (in blue), FERM-predictive (in red), and FERM-decorrelated (in green) for various unfairness levels $r$; lower values are better. Right panel: Unfairness levels on test data ($r_{test}$) for FRRM (in blue), FERM-predictive (in red), and FERM-decorrelated (in green) at varying unfairness levels $r$. Simulation settings are described in Table 1.

these components $X = [X_{\text{corr}}, X_{\text{indep}}]$. After generating $S$, $X$, and $Y$, all variables are standardized to have zero mean and unit variance to ensure consistency in the modeling process. The dataset is then split into training and testing sets, where $80\%$ of the samples are randomly assigned to the training set, and the remaining $20\%$ are assigned to the testing set. We generate synthetic datasets $\{(X_i, Y_i, S_i)\}_{i=1}^n$ under three settings, as listed in Table 1. We consider two types of noise distributions for $\eta$: in the normal setting, each element of $\eta$ follows a standard normal distribution, $\eta_{ij} \sim \mathcal{N}(0, 1)$, whereas in the Poisson setting, we set $\eta_{ij} \sim \text{Poi}(1) - 1$ (Results for Poisson setting (3) are provided in the Appendix E, Figure 9 ). The proposed FERM method and the baseline FRRM method are evaluated across varying unfairness constraints, defined by the unfairness budget $r \in \{0.1, 0.2, \ldots, 0.5\}$. Each simulation is repeated 50 times for every $r$ to account for variability in the results.

Across all settings, as shown in Figure 2, both decomposition-based models outperform FRRM in predictive accuracy (lower MSE) while maintaining comparable unfairness levels $r$, consistent with the theory in Section 4. **Fairness-utility trade-off (left panels):** at any fixed unfairness budget $r$, both FERM variants achieve lower test MSE than FRRM, with especially pronounced gains for FERM–predictive in higher-dimensional settings due to its removal of prediction-irrelevant components. **Attained unfairness (right panels):** all methods closely track the target unfairness budget on the test set, indicating that the fairness constraint is accurately enforced. In low dimensions, the two FERM variants perform similarly, but in high dimensions FERM-predictive is consistently superior, as its decomposition more effectively isolates the response-relevant directions in $X$. Both

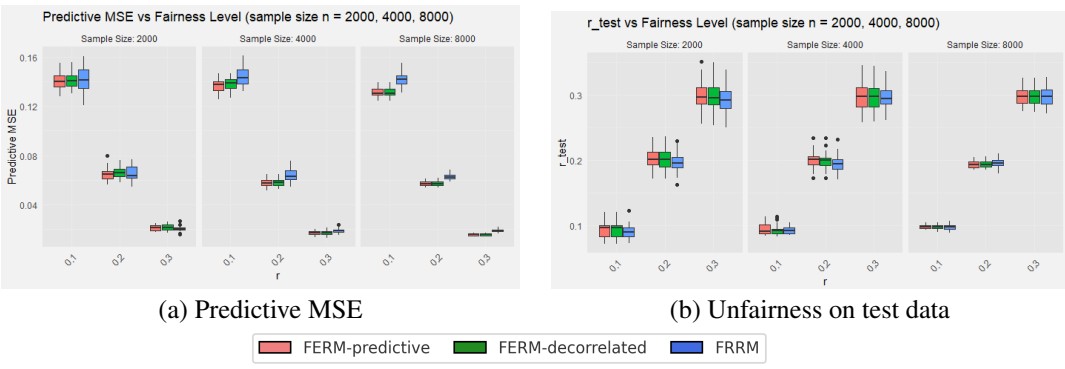

(a) Predictive MSE             (b) Unfairness on test data

Figure 3: Left panel: Predictive MSE for FRRM (in blue), FERM-predictive (in red), and FERM-decorrelated (in green) for unfairness levels $r = \{0.1, 0.2, 0.3\}$; lower values are better. Right panel: Unfairness levels on test data ($r_{test}$) for FRRM (in blue), FERM-predictive (in red), and FERM-decorrelated (in green) at unfairness levels $r = \{0.1, 0.2, 0.3\}$.

variants remain robust under non-Gaussian noise (see Figure 9). Overall, the simulations confirm that envelope-based decomposition improves statistical efficiency while preserving fairness.

## 6.1 REAL-WORLD DATA

For our real data application, we use U.S. Health Insurance Dataset (available at: https://www.kaggle.com/datasets/teertha/ushealthinsurancedataset). In this analysis, the sensitive attributes include gender, age, medical history, family medical history, and region. After creating dummy variables for all categorical variables, we finally have 15 features in $S$ and 15 features in $X$. We evaluated the performance under various settings with $r \in 0.1, 0.2, 0.3$ and sample sizes $n \in 2000, 4000, 8000$. The real data was perturbed and each configuration was replicated 30 times. The results are provided in the Figure 3. Results demonstrate that FERM models are superior to FRRM approach in most of the settings.

## 7 CONCLUSION

We introduced a new framework for fairness-aware regression that decomposes the predictor space into response-specific, sensitive, shared, and residual components. By penalizing only the sensitive subspaces, FERM provides an interpretable and tunable mechanism for balancing fairness and predictive accuracy. Our theoretical results establish efficiency gains relative to OLS and show that fairness can be enforced without discarding predictive signal. Empirical studies on both simulated and real-world datasets confirm that FERM consistently improves the fairness-utility trade-off compared to existing regression-based approaches. Beyond strong empirical performance, the key advantage of FERM lies in its interpretability: fairness constraints are imposed at the subspace level, making explicit how sensitive information enters the model. This transparency distinguishes FERM from black-box debiasing methods and offers practitioners a principled lever to manage fairness requirements. Nevertheless, FERM has important limitations. The method assumes linear subspace decompositions, and its fairness control is tied to covariance-based independence. We highlight these limitations in Appendix H, along with directions for extensions to nonlinear representations, alternative fairness notions, and scalable algorithms for high-dimensional predictors. More broadly, envelope methodology continues to expand. Extensions to generalized linear models (Cook & Zhang, 2015; Forzani & Su, 2021), matrix- and tensor-valued responses and predictors (Cook & Zhang, 2018; Li & Zhang, 2017), and alternative fairness penalties (Scutari et al., 2022) provide natural avenues to adapt and generalize the FERM framework. We discuss these directions, including integration with broader definitions of fairness, in Appendix C.

Overall, our results highlight predictor-space decomposition as a powerful tool for fairness-aware learning. We hope this work encourages further exploration of envelope methods at the intersection of statistical efficiency, interpretability, and algorithmic fairness.

## ETHICS STATEMENT

This work introduces FERM as a methodological tool to improve fairness and efficiency in regression. Simulations use only synthetic data, and the real-world experiment relies on a publicly available, anonymized Kaggle dataset. The method is designed to reduce unfair dependence on sensitive variables, though fairness guarantees remain context-dependent. This research complies with the ICLR Code of Ethics and has no conflicts of interest.

## REPRODUCIBILITY STATEMENT

We have made significant efforts to ensure reproducibility. All theoretical results are accompanied by formal proofs in Appendix B, with assumptions stated explicitly in Section 5. The full simulation setup, parameter choices, and evaluation protocol are described in Section 6 and Appendix F. Code to reproduce all synthetic experiments and figures, along with scripts for preprocessing the publicly available Kaggle dataset used in the real-world study, is included in the supplementary material. Random seeds and hyperparameter selection procedures are documented to enable exact replication.

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

# APPENDIX

## A  ENVELOPE REGRESSION MODELS

In this section, we illustrate the envelope regression methods, which aims to improve estimation efficiency by identifying relevant subspaces in the response or predictor space Cook et al. (2010; 2013), forming the foundation of our proposed methodology.

**Definition A.1** (M-Envelopes, (Cook et al., 2010))**.** For $M \in \mathbf{R}^{d \times d}$ and $\mathcal{B} \subseteq \mathrm{span}(M)$, the $M-$envelope of $\mathcal{B}$, denoted as $\mathcal{E}_M(\mathcal{B})$, is defined as the intersection of all reducing subspaces of $M$ that contains $\mathcal{B}$.

**Response Envelope Regression Model**   We employ the response envelope model, which assumes that a part of the predictors $\mathbf{X}$ remain stochastically constant as the sensitive attributes $\mathbf{S}$ vary. Consider the model:

$$X = B^\top S + e, \tag{11}$$

where $e$ is zero-mean normal noise with covariance matrix $Cov(e) = \Sigma_X$. The response envelope model assumes that the distribution of certain linear combinations of $X$ is invariant to $S$. This decomposition is formalized using two orthogonal matrices $\Gamma \in \mathbb{R}^{d_X \times m}$ and $\Gamma_0 \in \mathbb{R}^{d_X \times (d_X - m)}$, where $[\Gamma \ \Gamma_0]$ is an orthogonal matrix. Then we have:

(a) $\Gamma_0^\top X \mid S \sim \Gamma_0^\top X$, indicating invariance of $\Gamma_0^\top X$ to $S$.

(b) $\Gamma^\top X$ is uncorrelated with $\Gamma_0^\top X$ given $S$.

These conditions imply that $\Gamma_0^\top X$ carries no information about $S$. Furthermore, Cook et al. (2010) showed that conditions (a) and (b) are equivalent to:

(a') $\mathrm{span}(B) \subset \mathrm{span}(\Gamma)$.

(b') $\Sigma_X = \Sigma_1 + \Sigma_2 = P_\Gamma \Sigma_X P_\Gamma + Q_\Gamma \Sigma_X Q_\Gamma,$

where $\Sigma_X$ is the covariance of $X$, $P_\Gamma$ and $Q_\Gamma = I - P_\Gamma$ are projection operators onto $\mathrm{span}(\Gamma)$ and $\mathrm{span}(\Gamma_0)$, respectively. Based on these conditions, the response envelope model can be expressed as (Conway, 2019):

$$X = \Gamma \zeta S + e,$$
$$\Sigma_X = \Gamma \Omega \Gamma^\top + \Gamma_0 \Omega_0 \Gamma_0^\top, \tag{12}$$

where $\zeta \in \mathbf{R}^{m \times d_S}$ denotes the coefficients, the columns of $\Gamma$ is an orthogonal basis for the $\Sigma_X$-envelope of $\mathrm{span}(B)$, denoted by $\mathcal{E}_{\Sigma_X}(\mathrm{span}(B))$, and $m = \dim\big(\mathcal{E}_{\Sigma_X}(\mathrm{span}(B))\big)$. The matrices $\Omega \in \mathbb{R}^{m \times m}$ and $\Omega_0 \in \mathbb{R}^{(d_X - m) \times (d_X - m)}$ provide the coordinates of $\Sigma_X$ with respect to $\Gamma$.

The visual representation (Figure 4) illustrates this decomposition for $X \in \mathbb{R}^2$ and $S \in \mathbb{R}$. Each point represents an observation $X = (X_1, X_2)$, colored according to its value of the sensitive attribute $S$.

- The blue arrow, labeled $\mathrm{Span}(\Gamma)$, indicates the direction in the $X-$space along which the values of $X$ change significantly with $S$. Observations with similar $S$ values tend to cluster or vary predominantly along this direction.

- The red arrow, labeled $\mathrm{Span}(\Gamma_0)$, represents the direction in the $X-$space along which the values of $\mathbf{X}$ are invariant of $S$. Variations along this direction show no systematic change with the sensitive attribute.

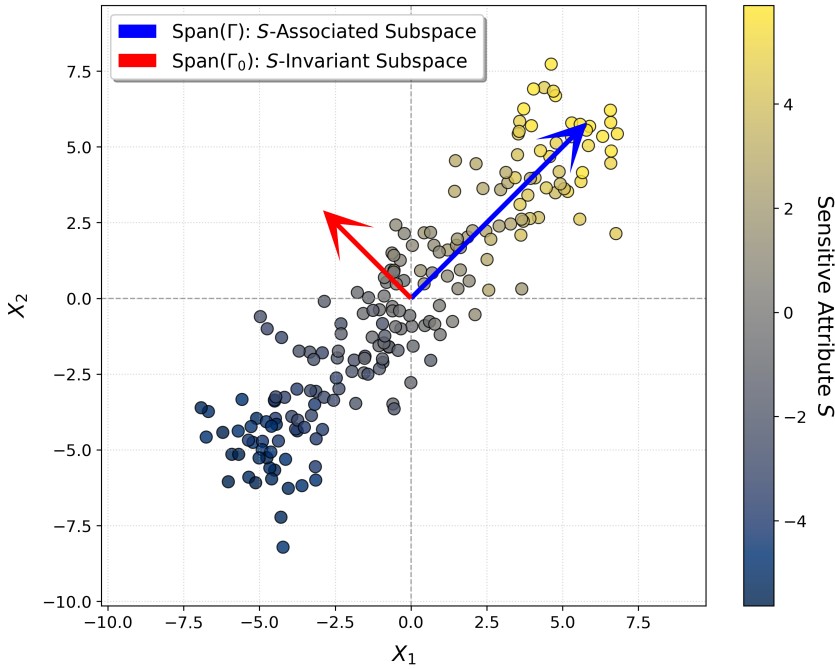

Figure 4: Response $X$ with Sensitive Attribute $S$ and Envelope Subspaces. The data points are colored by the value of $S$. The blue arrow indicates the direction of the sensitive-associated subspace Span($\Gamma$), while the red arrow indicates the sensitive-invariant subspace Span($\Gamma_0$).

**Predictor Envelope Regression Model**  The predictor envelope model Cook et al. (2013) identifies material and immaterial parts of the predictors $X$ in the regression of $Y$ on $X$. The model is defined as:

$$Y = \mu_Y + \beta^\top (X - \mu_X) + \epsilon, \tag{13}$$

where $\epsilon$ is zero-mean noise. The predictor envelope method seeks a dimension reduction for $X$ by finding the $\Sigma_X$-envelope of span($\beta^\top$), denoted by $\varepsilon_{\Sigma_X}(\text{span}(\beta^\top))$. This decomposition divides $X$ into material and immaterial parts, denotes as $P_M X$ and $Q_M X$, which satisfies

(i) $Q_M X$ and $P_M X$ are uncorrelated.

(ii) $Q_M X$ is uncorrelated with $Y$ given $P_M X$.

Let $u = \dim\left(\varepsilon_{\Sigma_X}(\text{span}(\beta^\top))\right)$, and let $\Phi \in \mathbb{R}^{p \times u}$ and $\Phi_0 \in \mathbb{R}^{p \times (p-u)}$ be orthogonal bases for $\varepsilon_{\Sigma_X}(\text{span}(\beta^\top))$ and its orthogonal complement, respectively. The predictor envelope model can then be written as:

$$Y = \mu_Y + \psi^\top \Phi^\top (X - \mu_X) + \epsilon,$$
$$\Sigma_X = \Phi \Delta \Phi^\top + \Phi_0 \Delta_0 \Phi_0^\top, \tag{14}$$

where $\beta = \Phi\psi$, and $\Delta \in \mathbb{R}^{u \times u}$ and $\Delta_0 \in \mathbb{R}^{(p-u) \times (p-u)}$ provide the coordinates of $\Sigma_X$ with respect to $\Phi$.

*Remark* A.2. Throughout the paper, we use the terms "associated" for $S$ and "predictive" for $Y$ because the two envelopes play fundamentally different roles in our framework. The matrix $\Gamma$ spans directions of $X$ that are *associated* with $S$, that is, the variation in $X$ that can be explained by $S$. In contrast, $\Phi$ spans the directions of $X$ that are *predictive* of $Y$, i.e., the variation that is material for the regression of $Y$ on $X$. Their complements naturally take on different names: $\Gamma_0$ corresponds to $S$-invariant directions, while $\Phi_0$ corresponds to $Y$-immaterial directions. This terminology is directly inherited from the envelope regression literature, where predictor envelopes (for $S$) and response envelopes (for $Y$) describe these two distinct types of material–immaterial decompositions.

## A.1 ENVELOPE ESTIMATION OBJECTIVE

Given the dimension $u$ of an envelope subspace, envelope estimation reduces to solving the constrained optimization problem

$$\hat{\Gamma} = \arg \min_{\Gamma \in \mathbb{R}^{p \times u} \,:\, \Gamma^\top \Gamma = I_u} J_n(\Gamma), \qquad J_n(\Gamma) = \log \left| \Gamma^\top \hat{M} \Gamma \right| + \log \left| \Gamma^\top (\hat{M} + \hat{U})^{-1} \Gamma \right|, \quad (15)$$

where $\hat{M} \succ 0$ and $\hat{U} \succeq 0$ are finite-sample estimators of the population matrices $M$ and $U$. The envelope estimator is then defined as

$$\widehat{\mathcal{E}}_M(U) = \text{span}(\hat{\Gamma}).$$

The constraint $\Gamma^\top \Gamma = I_u$ ensures that $\Gamma$ has orthonormal columns. Consequently, optimization problem (15) is *non-convex* and takes place on a Stiefel manifold. If we treat the subspace $\mathcal{S} = \text{span}(\Gamma)$ as the argument rather than the basis matrix $\Gamma$, then the problem becomes equivalent to a non-convex optimization over the Grassmann manifold (the set of $u$-dimensional subspaces of $\mathbb{R}^p$).

This formulation highlights that envelope estimation is inherently a problem of manifold optimization. Almost all envelope methods, including those used in regression, prediction, and our fairness-aware adaptation (FERM), are connected through this shared objective structure.

**Choice of $\hat{M}$ and $\hat{U}$:**   The specific form of $\hat{M}$ and $\hat{U}$ depends on whether one is estimating a *response envelope* or a *predictor envelope*. Let $\hat{\Sigma}_Y$, $\hat{\Sigma}_X$, $\hat{\Sigma}_S$, $\hat{\Sigma}_{XY}$, $\hat{\Sigma}_{XS}$ and $\hat{\Sigma}_{YX} = \hat{\Sigma}_{XY}^\top$, $\hat{\Sigma}_{SX} = \hat{\Sigma}_{XS}^\top$ denote the (centered) sample covariance and cross-covariance matrices computed from $\{(X_i, S_i, Y_i)\}_{i=1}^n$. As described in Cook & Zhang (2018); Zhang et al. (2023), Then the empirical counterparts used in equation 15 are:

- Response envelope model in (12) :

$$\hat{M} = \hat{\Sigma}_X, \qquad \hat{U} = \hat{\Sigma}_{XS} \hat{\Sigma}_{SX}.$$

- Predictor envelope model in (14) :

$$\hat{M} = \hat{\Sigma}_X, \qquad \hat{U} = \hat{\Sigma}_{XY} \hat{\Sigma}_{YX}.$$

# B   PROOFS OF RESULTS IN SECTION 5

**Proof of Theorem 5.1:**

Recall that $\hat{\beta}_{\text{env}} := P \hat{\beta}_{\text{OLS}}$, where $P$ is the orthogonal projector onto the subspace spanned by $\Pi_{XSY}$ and $\Pi_{XY}$. Let $\Theta \in \mathbb{R}^{d_X \times k}$ have orthonormal columns forming a basis for this subspace, and let $\Theta_0 \in \mathbb{R}^{d_X \times (d_X - k)}$ form an orthonormal basis for its orthogonal complement, where $k = \dim(\text{span}\{\Pi_{XSY}, \Pi_{XY}\})$.

Since $\beta \in \text{range}(P)$, we have $P\beta = \beta$. Thus,

$$\sqrt{n}\,(\hat{\beta}_{\text{env}} - \beta) = \sqrt{n}\, P(\hat{\beta}_{\text{OLS}} - \beta).$$

Because $v \mapsto Pv$ is linear and continuous, the continuous mapping theorem combined with the assumed asymptotic normality of $\hat{\beta}_{\text{OLS}}$ gives

$$\sqrt{n}\,(\hat{\beta}_{\text{env}} - \beta) \xrightarrow{d} N\big(0,\, P\,\Sigma\,P\big),$$

where $\Sigma := avar(\sqrt{n}\,\hat{\beta}_{\text{OLS}})$. Therefore,

$$avar(\sqrt{n}\,\hat{\beta}_{\text{env}}) = P\Sigma P.$$

Next, express $P$ and $I - P$ in the chosen orthonormal basis:

$$P = \Theta\Theta^\top, \qquad I - P = \Theta_0\Theta_0^\top.$$

With respect to this basis, $\Sigma$ has the block form

$$\Sigma = [\Theta \quad \Theta_0] \begin{bmatrix} \Sigma_{11} & \Sigma_{12} \\ \Sigma_{21} & \Sigma_{22} \end{bmatrix} [\Theta \quad \Theta_0]^\top, \quad \text{where } \Sigma_{ij} = \Theta_i^\top \Sigma \Theta_j.$$

By construction of the envelope subspace, $\Sigma_{12} = \Sigma_{21}^\top = 0$, so $\Sigma$ is block-diagonal with respect to $P \oplus P^\perp$. Hence,

$$P\Sigma P = \Theta\,\Sigma_{11}\,\Theta^\top, \qquad \Sigma - P\Sigma P = \Theta_0\,\Sigma_{22}\,\Theta_0^\top.$$

Since $\Sigma \succeq 0$ implies $\Sigma_{22} \succeq 0$, we obtain $\Sigma - P\Sigma P \succeq 0$, i.e.,

$$P\Sigma P \preceq \Sigma.$$

Finally, if $\Theta_0$ is nontrivial and $\Sigma_{22} \succ 0$, then

$$\Sigma - P\Sigma P = \Theta_0\,\Sigma_{22}\,\Theta_0^\top \succ 0,$$

which establishes the strict inequality $P\Sigma P \prec \Sigma$. $\qquad\qquad\qquad\qquad\square$

**Proof of Proposition 5.2.** The result follows by combining the asymptotic normality of $\hat\beta_{\text{OLS}}$ with the $\sqrt{n}$-consistency of envelope subspace estimators.

First, by standard linear model theory,

$$\sqrt{n}(\hat\beta_{\text{OLS}} - \beta) \xrightarrow{d} N(0, \Sigma),$$

so $\hat\beta_{\text{OLS}}$ is $\sqrt{n}$-consistent for $\beta$.

Second, Algorithm 1 estimates the envelope projector $\hat{P}$ using either the 1D algorithm of Cook & Zhang (2016) or the NIECE algorithm of Zhang et al. (2023). Proposition 6 in Cook & Zhang (2016) and Theorem 1 in Zhang et al. (2023) both establish that the estimated envelope subspace converges at $\sqrt{n}$-rate to the population envelope subspace, i.e.,

$$\|\hat{P} - P\| = O_p(n^{-1/2}).$$

Now decompose

$$\hat\beta_{\text{env}}(\hat{P}) - \beta = (\hat{P} - P)\hat\beta_{\text{OLS}} + P(\hat\beta_{\text{OLS}} - \beta).$$

Since $\hat\beta_{\text{OLS}} = O_p(1)$ and $\hat{P} - P = O_p(n^{-1/2})$. The second term is asymptotically normal with mean zero and covariance $P\Sigma P$ by Theorem 5.1.

Therefore,

$$\sqrt{n}\big(\hat\beta_{\text{env}}(\hat{P}) - \beta\big) = O_p(1),$$

showing that $\hat\beta_{\text{env}}$ is $\sqrt{n}$-consistent. $\qquad\qquad\qquad\qquad\square$

**Proof of Theorem 5.4:**

Note that

$$\hat\beta_{XY}^{\text{fair}} = \arg\min_{\beta_{XY}} \|\mathbf{Y} - \mathbf{X}_{XY}\beta_{XY}\|_2^2,$$

$$(\hat\beta_{XY}^{\text{OLS}}, \hat\beta_{XSY}^{\text{OLS}}) = \arg\min_{\beta_{XY}, \beta_{XSY}} \|\mathbf{Y} - \mathbf{X}_{XY}\beta_{XY} - \mathbf{X}_{XSY}\beta_{XSY}\|_2^2,$$

$$(\hat\beta_{XY}(\lambda), \hat\beta_{XSY}(\lambda)) = \arg\min_{\beta_{XY}, \beta_{XSY}} \|Y - X_{XY}\beta_{XY} - X_{XSY}\beta_{XSY}\|_2^2 + \lambda\|\beta_{XSY}\|_2^2,$$

with penalty $\lambda \geq 0$ applied only to the shared component. Thus,

$$\hat\beta_{XY}(\lambda) = \hat\beta_{XY}^{\text{OLS}} = \left(\hat\Pi_{XY}^\top \mathbf{X}^\top \mathbf{X} \hat\Pi_{XY}\right)^{-1} \hat\Pi_{XY}^\top \mathbf{X}^\top \mathbf{Y},$$

$$\hat\beta_{XSY}(\lambda) = \left(\mathbf{X}_{XSY}^\top \mathbf{X}_{XSY} + \lambda(r)I\right)^{-1} \mathbf{X}_{XSY}^\top \mathbf{Y}.$$

Using Sherman-Morrison-Woodbury identity for matrices, we have

$$\left(\mathbf{X}_{XSY}^{\top}\mathbf{X}_{XSY} + \lambda(r)I\right)^{-1} = \left(\mathbf{X}_{XSY}^{\top}\mathbf{X}_{XSY}\right)^{-1} - \left(\mathbf{X}_{XSY}^{\top}\mathbf{X}_{XSY}\right)^{-1}\left(\frac{1}{\lambda(r)}\mathbf{X}_{XSY}^{\top}\mathbf{X}_{XSY} + I\right)^{-1}.$$

Thus,

$$\hat{\beta}_{XSY}(\lambda) = \left(\mathbf{X}_{XSY}^{\top}\mathbf{X}_{XSY}\right)^{-1}\mathbf{X}_{XSY}^{\top}\mathbf{Y} - \left(\mathbf{X}_{XSY}^{\top}\mathbf{X}_{XSY}\right)^{-1}\left(\frac{1}{\lambda(r)}\mathbf{X}_{XSY}^{\top}\mathbf{X}_{XSY} + I\right)^{-1}\mathbf{X}_{XSY}^{\top}\mathbf{Y}$$

$$= \hat{\beta}_{XSY}^{\mathrm{OLS}} - \left(\mathbf{X}_{XSY}^{\top}\mathbf{X}_{XSY}\right)^{-1}\left(\frac{1}{\lambda(r)}\mathbf{X}_{XSY}^{\top}\mathbf{X}_{XSY} + I\right)^{-1}\mathbf{X}_{XSY}^{\top}\mathbf{Y}.$$

This allows us to obtain the following representation, using $\mathbf{X}_{XSY}^{\top}\mathbf{X}_{XSY} = nI$,

$$\hat{Y}_{\lambda}(X, S) = \hat{\beta}_{XY}(\lambda)^{\top}X_{XY} + \hat{\beta}_{XSY}^{\top}(\lambda)X_{XSY}$$

$$= (\hat{\beta}_{XY}^{\mathrm{OLS}})^{\top}X_{XY} + (\hat{\beta}_{XSY}^{\mathrm{OLS}})^{\top}X_{XSY} - ((\mathbf{X}_{XSY}^{\top}\mathbf{X}_{XSY})^{-1}\left(\frac{1}{\lambda(r)}\mathbf{X}_{XSY}^{\top}\mathbf{X}_{XSY} + I\right)^{-1}\mathbf{X}_{XSY}^{\top}\mathbf{Y})^{\top}X_{XSY}$$

$$= (\hat{\beta}_{XY}^{\mathrm{OLS}})^{\top}X_{XY} + (\hat{\beta}_{XSY}^{\mathrm{OLS}})^{\top}X_{XSY} - \frac{1}{n}\frac{\lambda(r)}{n + \lambda(r)}(\mathbf{X}_{XSY}^{\top}\mathbf{Y})^{\top}X_{XSY}$$

$$= (\hat{\beta}_{XY}^{\mathrm{OLS}})^{\top}X_{XY} + (\hat{\beta}_{XSY}^{\mathrm{OLS}})^{\top}X_{XSY} - \frac{\lambda(r)}{n + \lambda(r)}(\hat{\beta}_{XSY}^{\mathrm{OLS}})^{\top}X_{XSY}$$

On rearranging, we get

$$\hat{Y}_{\lambda} = \frac{n}{n + \lambda(r)}\hat{Y}_{\mathrm{OLS}} + \left(1 - \frac{n}{n + \lambda(r)}\right)\hat{Y}_{\mathrm{fair}}, \qquad \lambda(r) \geq 0$$

Let $\omega(n, r) = \frac{n}{n+\lambda(r)}$. Thus,

$$Y - \hat{Y}_{\lambda} = Y - \left[\omega(n, r)\hat{Y}_{\mathrm{OLS}} + (1 - \omega(n, r))\hat{Y}_{\mathrm{fair}}\right] = (Y - \hat{Y}_{\mathrm{OLS}}) + (1 - \omega(n, r))(\hat{Y}_{\mathrm{OLS}} - \hat{Y}_{\mathrm{fair}})$$

Hence

$$E\left[Y - \hat{Y}_{\lambda}\right]^2 = E\left[Y - \hat{Y}_{\mathrm{OLS}}\right]^2 + (1 - \omega(n, r))^2 E\left[\hat{Y}_{\mathrm{OLS}} - \hat{Y}_{\mathrm{fair}}\right]^2$$
$$+ 2(1 - \omega(n, r))E\left[(Y - \hat{Y}_{\mathrm{OLS}})(\hat{Y}_{\mathrm{OLS}} - \hat{Y}_{\mathrm{fair}})\right].$$

But note that $(Y - \hat{Y}_{\mathrm{OLS}})$ and $(\hat{Y}_{\mathrm{OLS}} - \hat{Y}_{\mathrm{fair}})$ are orthogonal, hence uncorrelated. Since, $E(e) = 0$, the cross-term above vanishes.

That is,

$$\mathrm{MSE}_{\lambda} = \mathrm{MSE}_{\mathrm{OLS}} + (1 - \omega(n, r))^2 \mathrm{MSE}_{\mathrm{fair}}.$$

showing exactly how FERM interpolates between the two models via the tuning parameter $\lambda(r)$.

## C WHEN ARE LINEAR ENVELOPES APPROPRIATE? NONLINEAR GENERALIZATIONS OF FERM

The FERM framework is built on an explicit four-way linear decomposition of the predictor space into orthogonal components capturing predictive, sensitive, shared, and residual variation. This decomposition - combined with the closed-form characterization of the fairness–utility trade-off - is

only achievable under a linear model for $Y$. Linear envelopes therefore serve as the mathematically tractable foundation on which our theoretical guarantees rest.

This structure mirrors the classical development of PCA: the linear PCA formulation provides interpretability, closed-form solutions, and a clear geometric foundation; nonlinear variants such as kernel PCA arise only after the linear theory is firmly established. Our contribution plays an analogous role in fairness-aware regression: it formalizes the linear envelope setting as a principled, interpretable baseline with guarantees on efficiency and fairness.

Nevertheless, linear envelopes are not the end point. FERM is constructed with a modular and flexible design, which enables a wide range of possible extensions. These include incorporating nonlinear modeling frameworks, introducing alternative or more sophisticated penalization schemes, applying different definitions of fairness, and adapting the method to various types of response variables using generalized linear models. A key strength of FERM lies in the clear separation between model selection (i.e., the choice of $\lambda(r)$) and model estimation, which facilitates independent modification of either stage. This separation allows FERM to draw from and integrate with a broad body of established techniques in statistical modeling, offering adaptability to different fairness-aware regression settings.

## C.1 Nonlinear Regression Models

FERM can be extended to handle nonlinear relationships by incorporating kernel methods, in a similar fashion as suggested by Komiyama et al. (2018) and Scutari et al. (2022). Specifically, the model can be fitted into transformed feature spaces $Z_\Gamma(\hat{\Gamma}^T X)$ and $Z_{\Gamma_0}(\hat{\Gamma}_0^T X)$, obtained through positive definite kernel functions, similar to the approach in Komiyama et al. (2018). Applying the kernel trick in conjunction with ridge penalization leads to a kernel ridge regression variant of FERM (cf. Saunders et al. (1998)), which can be estimated efficiently using techniques such as those proposed by Zhang et al. (2015). Moreover, since kernel ridge regression is closely related to Gaussian process regression, this extension naturally opens the door to Bayesian nonparametric variants of FERM using Gaussian processes, as discussed in Kanagawa et al. (2018).

## C.2 Extensions to Deep Learning Models

The FERM framework can also be extended to deep learning architectures by leveraging the envelope decomposition as a structured preprocessing step. Specifically, the decomposition of the predictor space into orthogonal components – associated with the response ($Y$), the sensitive attribute ($S$), their shared variation, and residuals – provides a principled way to separate and control different sources of variation prior to feeding them into a neural network. One approach is to project the raw inputs onto the learned subspaces, and then use only the subspace orthogonal to the sensitive attributes as input to the downstream model. This can help ensure that the representations learned by the network are less entangled with sensitive information, improving fairness in predictions.

Taken together, the linear envelope provides the theoretical scaffold, while the nonlinear extensions offer paths toward greater flexibility. This Appendix clarifies when the linear formulation is appropriate (e.g., when interpretability, closed-form decomposition, or theoretical guarantees are desired) and how the nonlinear generalizations can be systematically constructed.

## C.3 Different Definitions of Fairness

The modular structure of FERM allows flexibility in how fairness is defined and enforced. In particular, the fairness constraint used in the form $R^2(\boldsymbol{\alpha}, \boldsymbol{\beta}) \leq r$ in our methodology can be modified independently of the estimation procedure for $\boldsymbol{\alpha}_{\text{FERM}}$ and $\boldsymbol{\beta}_{\text{FERM}}$. In particular, the extension reported in Section 4.3 of Scutari et al. (2022) can be directly applied to achieve this in our case. For instance, this constraint can be replaced by an analogous constraint based on *equality of opportunity*. One such measure is:

$$R_{\text{EO}}^2(\boldsymbol{\phi}, \boldsymbol{\psi}) = \frac{\text{Var}(S\boldsymbol{\phi})}{\text{Var}(Y\boldsymbol{\psi} + S\boldsymbol{\phi})},$$

where $\phi$, $\psi$ are the regression coefficients in the model $\hat{Y}_\lambda = Y\psi + S\phi + \varepsilon^*$ and $\hat{Y}_\lambda$ as defined before. If equality of opportunity holds exactly, then $\hat{Y}_{\text{FERM}}$ is conditionally independent of $S$ given $Y$, i.e., $\text{Cov}(\hat{Y}_\lambda, S \mid Y) = 0$. This implies $\phi = \mathbf{0}$ and $R^2_{\text{EO}} = 0$. FERM can approximate this condition asymptotically: as $\lambda(r) \to \infty$, we have $\hat{Y}_\lambda \to \hat{Y}_{\text{fair}}$, leading to vanishing conditional covariance. Conversely, as $\lambda(r) \to 0$, the fairness constraint becomes inactive, and $\hat{Y}_\lambda \to \hat{Y}_{\text{OLS}}$. For finite $\lambda(r)$, we obtain $\hat{Y}_\lambda = \omega(n, r)\,\hat{Y}_{\text{OLS}} + \left(1 - \omega(n, r)\right)\hat{Y}_{\text{fair}}$, implying that $\text{Cov}(\hat{Y}_\lambda, S \mid Y)$ and thus $R^2_{\text{EO}}$ decrease as $\lambda(r)$ increases. This mirrors the control we exert over $R^2$ in Section 4.

# D    ADDITIONAL RESULTS

In this section, we present additional simulation results to support our theoretical guarantees. We compare three regression settings:

$$\mathcal{M}_1 : Y \sim X_{XY} + X_{XSY} \qquad (\text{Env\_Xy\_Xys}),$$
$$\mathcal{M}_2 : Y \sim X_{XY} + X_0 + X_{XSY} \qquad (\text{Env\_XyX0\_Xys}),$$
$$\mathcal{M}_3 : Y \sim \underbrace{X_{XY} + X_0}_{U} + \underbrace{X_{XSY} + X_{XS}}_{S} \qquad (\text{FRRM}),$$

where "$\sim$" indicates that $Y$ is regressed on the specified terms.

Our evaluation focuses on three key objectives:

- **Coefficient convergence in the unconstrained case.** When $r = 1$, Theorem 5.4 shows that both envelope estimators ($\mathcal{M}_1$ and $\mathcal{M}_2$) converge to the ordinary least-squares estimator (FRRM in $\mathcal{M}_3$). We verify this by computing the $L_2$ distance between the envelope coefficients and the FRRM coefficients.

- **Fairness: convergence of covariance to zero.** When $r = 0$, Lemma 5.3 implies that the covariance between predictions and the sensitive component converges to zero in probability for all three models. We evaluate this by computing $\text{Cov}(\hat{Y}, S)$ on held-out test data.

- **Efficiency gain through subspace reduction.** The envelope model removes the redundant component $X_0$, which has no impact on $Y$. Eliminating this noise-inflating subspace reduces the variance of the estimated coefficients, as shown in Theorem 5.1, demonstrating the efficiency benefit of envelope-based dimension reduction.

We conduct experiments under two settings:

$$d_X = 30, \qquad d_S = 20, \qquad \dim(\Pi_{XY}) = \dim(\Pi_{XSY}) = \dim(\Pi_{XS}) = 5,$$
$$d_X = 100, \qquad d_S = 20, \qquad \dim(\Pi_{XY}) = \dim(\Pi_{XSY}) = \dim(\Pi_{XS}) = 10.$$

For each setting, we vary the sample size

$$n \in \{200, 500, 1000, 2000, 5000, 10000, 20000\},$$

and the unfairness control level

$$r \in \{0, 0.1, 0.3, 0.5, 0.7, 0.9, 1\}.$$

All configurations are replicated $50$ times. The corresponding results are summarized below.

From Figure 5, we observe that in both settings ($d_X = 30$ and $d_X = 100$) and in the unconstrained case ($r = 1$), the estimated coefficients from $\mathcal{M}_1$ and $\mathcal{M}_2$ converge toward those of $\mathcal{M}_3$, confirming the consistency result established in Theorem 5.4.

Figures 6 and 7 further show that when $r = 0$, all three models yield fully fair estimators: the covariance between the predicted value and the sensitive component decreases steadily as the sample size increases, in accordance with Lemma 5.3. In contrast, this covariance does not vanish when $r = 0.5$, since the model is not required to be fair and the predictions may legitimately depend on the sensitive direction.

Finally, Figure 8 demonstrates the efficiency gain obtained from removing the redundant subspace $X_0$. Across all dimensions and all fairness levels, the average variance ratio between $\mathcal{M}_2$ (which includes $X_0$) and $\mathcal{M}_1$ (which excludes $X_0$) is consistently greater than one. This confirms that eliminating components that do not contribute to the response reduces estimator variance and yields a more efficient model.

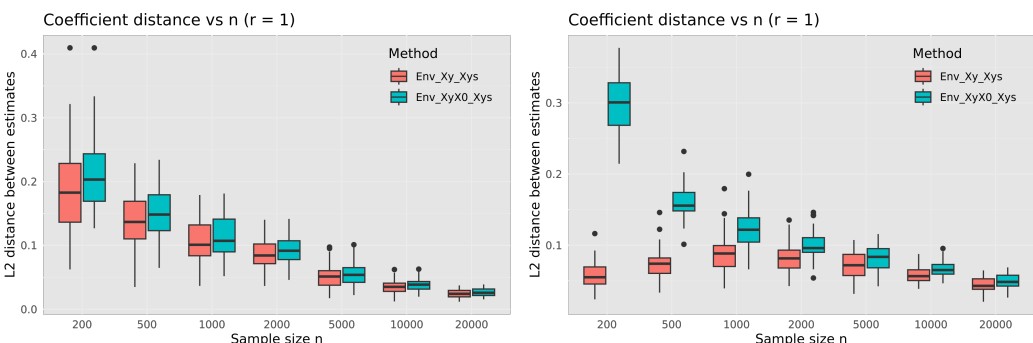

Figure 5: **Coefficient convergence under the unconstrained case** ($r = 1$). The plots show the $L_2$ distance between the estimated coefficients of the two envelope models ($\mathcal{M}_1$ and $\mathcal{M}_2$) and the FRRM estimator $\mathcal{M}_3$ across different sample sizes, under two dimensional settings ($d_X = 30$ and $d_X = 100$).

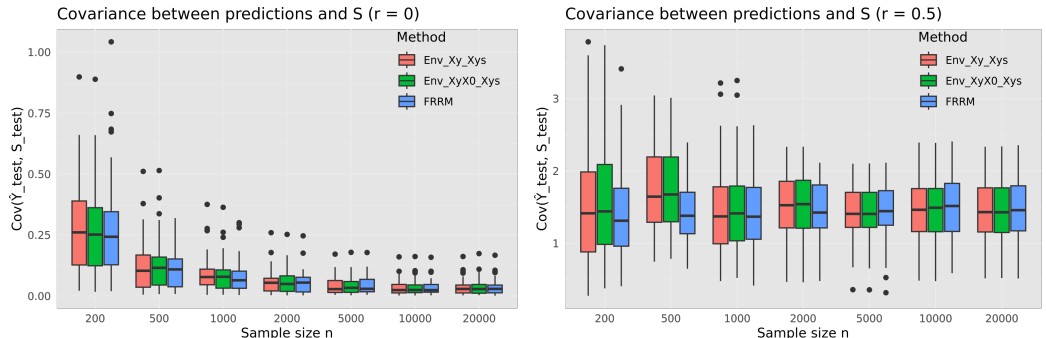

Figure 6: **Covariance between predictions and the sensitive component when** $d_X = 30$.
Left: When $r = 0$, all three models produce fully fair estimators, and the empirical covariance $\mathrm{Cov}(\widehat{Y}, S)$ decreases toward zero as the sample size increases. Right: When $r = 0.5$, the covariance does not converge to zero since the model is only partially fair.

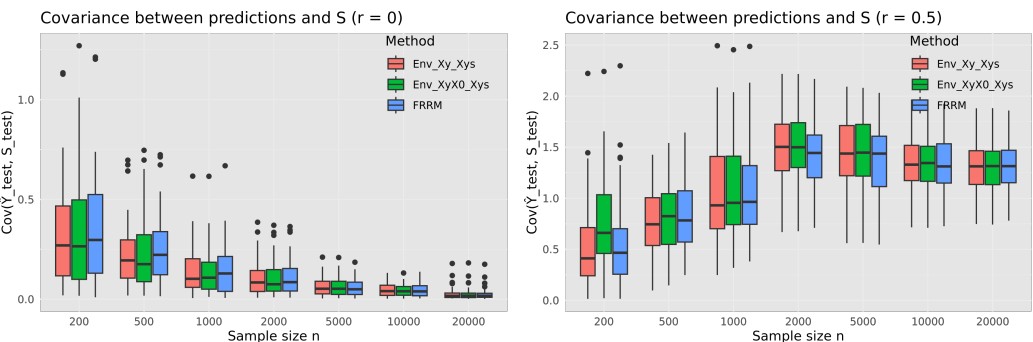

Figure 7: **Covariance between predictions and the sensitive component when** $d_X = 100$.

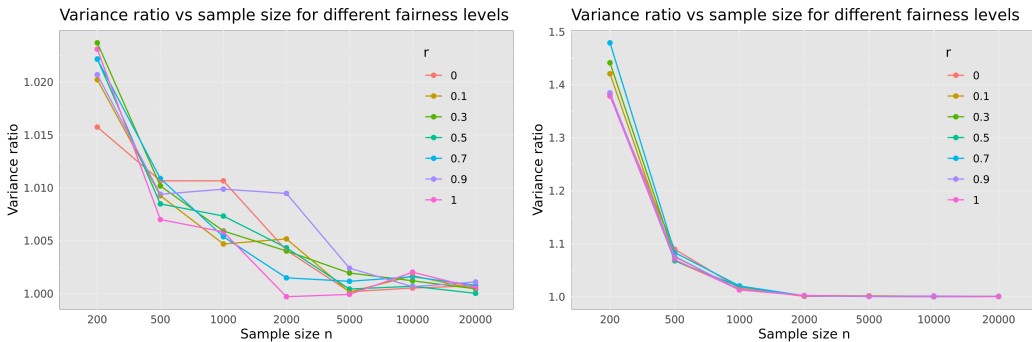

Figure 8: **Efficiency comparison via variance ratios of envelope estimators.** For each dimension setting ($d_X = 30$ (left panel) and $d_X = 100$ (right panel)), the plots show the ratio of the average coordinatewise variance of the estimator from $\mathcal{M}_2$ (which includes the redundant subspace $X_0$) to that from $\mathcal{M}_1$ (which excludes $X_0$). Across all fairness levels and sample sizes, this ratio is consistently greater than 1.

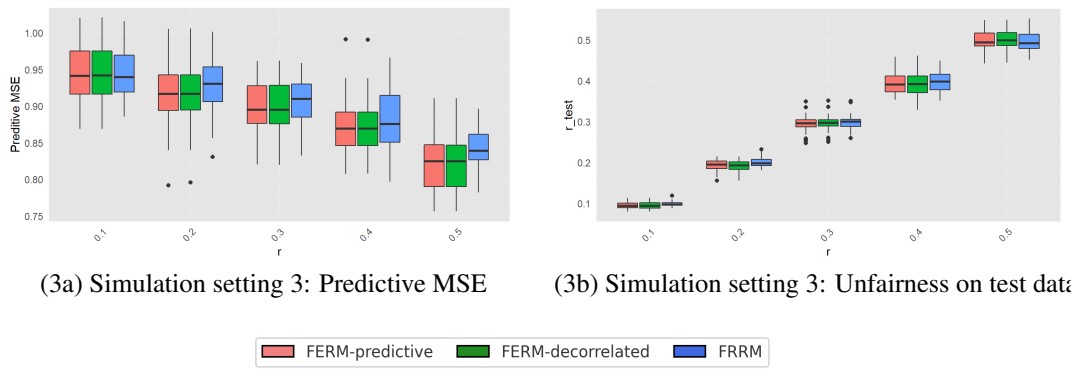

(3a) Simulation setting 3: Predictive MSE     (3b) Simulation setting 3: Unfairness on test data

Figure 9: Left panel: Predictive MSE for FRRM (in blue), FERM-predictive (in red), and FERM-decorrelated (in green) for various unfairness levels $r$; lower values are better. Right panel: Unfairness levels on test data ($r_{test}$) for FRRM (in blue), FERM-predictive (in red), and FERM-decorrelated (in green) at varying unfairness levels $r$. Simulation settings are descrsibed in Table 1.

# E EXPERIMENTS

## E.1 POISSON SETTING

In Figure 9,we provide simulation results for the Poisson setting (3) given in Table 1.

## E.2 ADDITIONAL REAL-WORLD DATA

In another real-data application, we use the Full-Year Consolidated (FYC) data files from the *Medical Expenditure Panel Survey (MEPS)* dataset (available at: `https://meps.ahrq.gov/data_stats/download_data_files.jsp`), which combine person-level demographics, socioeconomic status, insurance coverage, and all medical expenditures across the entire year. In this analysis, the outcome $Y$ is the total annual medical expenditure (`totexp`), treated as a continuous centered variable. The sensitive attributes $S$ consist of eight standardized socioeconomic and payment-related variables: age (`age`), family income (`faminc`), poverty level (`povlev`), and total payments from self-pay (`totself`), Medicare (`totmcr`), Medicaid (`totmcd`), and private insurance (`totprv`). The predictor set $X$ includes 100 continuous features derived from MEPS health, demographic, income, and expenditure variables after removing identifiers, low-variance fields, and

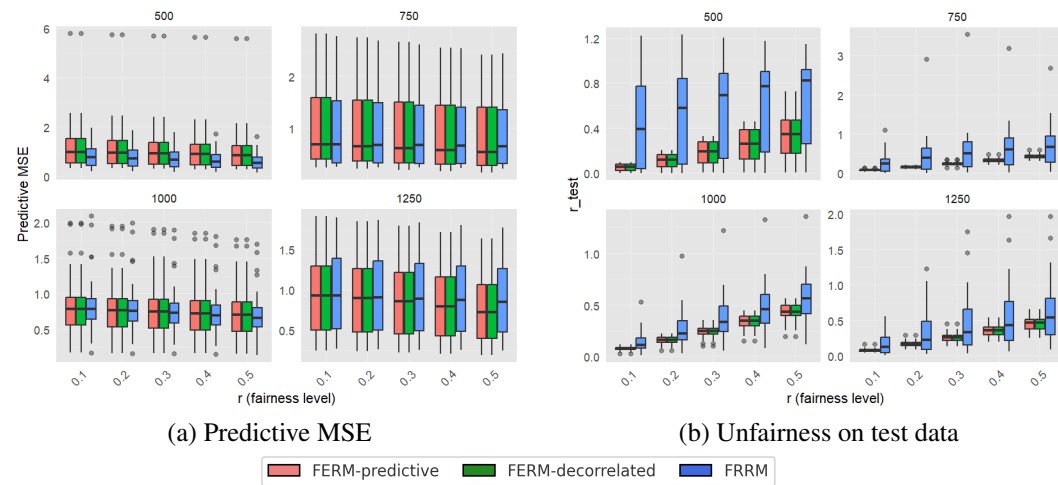

(a) Predictive MSE                    (b) Unfairness on test data

| FERM-predictive | FERM-decorrelated | FRRM |

Figure 10: Left panel: Predictive MSE for FRRM (in blue), FERM-predictive (in red), and FERM-decorrelated (in green) for unfairness levels $r = \{0.1, 0.2, 0.3, 0.4, 0.5\}$; lower values are better. Right panel: Unfairness levels on test data ($r_{test}$) for FRRM (in blue), FERM-predictive (in red), and FERM-decorrelated (in green) at unfairness levels $r = \{0.1, 0.2, 0.3, 0.4, 0.5\}$. The plots for sample sizes $n = 500, 750, 1000, 1250$ are shown here.

categorical-like columns.
We evaluate performance across fairness levels $r \in \{0.1, 0.2, 0.3, 0.4, 0.5\}$ and sample sizes $n \in \{500, 750, 1000, 1250\}$. The dataset is subsampled and perturbed, and each configuration is replicated 30 times. The corresponding results are shown in Figure 10. Across most settings, the FERM models outperform the FRRM baseline, demonstrating improved predictive efficiency under fairness constraints in this complex real-world healthcare environment.

**Summary of Results.** Across most configurations, the predictive performance of FERM is comparable to that of FRRM, as shown in the left panels of Figure 10. However, the right panels reveal a key distinction: FRRM often exceeds the target unfairness levels $r$, whereas FERM consistently adheres to the specified fairness constraint. This deviation explains the apparent similarity in predictive error - FRRM achieves lower error in some settings only by violating the intended fairness budget. In contrast, FERM provides more reliable and controlled fairness–utility trade-offs, maintaining performance while respecting the target unfairness levels.

### E.3    REPRODUCIBILITY

All code used to generate the simulation results in Section 6 is provided in the supplementary material. The repository includes data generation scripts, training routines for FERM and FRRM, and plotting code to reproduce all figures and tables. Random seeds and parameter settings are fixed to ensure exact replication.

### E.4    COMPUTATIONAL DETAILS:

We conduct all our experiments on a Gentoo Linux server with an Intel Xeon E5-2683 v4 @ 2.10GHz CPU and 251GB of RAM. No GPU is involved.

# F    IMPLEMENTATION DETAILS OF ALGORITHMS

## F.1    BASELINE IMPLEMENTATION DETAILS: FAIR RIDGE REGRESSION MODEL (FRRM)

For the Fair Ridge Regression Model (FRRM; Scutari et al. (2022)), we follow the authors' implementation exactly as recommended. All FRRM results were obtained using the `frrm()` function from the `fairml` package in R, with default settings unless otherwise specified. FRRM requires the user to specify a target unfairness budget $r$, and the algorithm internally determines the corresponding ridge penalty $\lambda$ to satisfy this constraint. No additional hyperparameter tuning is exposed to the user. In all our experiments (Section 6), we evaluated FRRM at unfairness levels $r = \{0.1, 0.2, 0.3, 0.4, 0.5\}$. For reproducibility, we ensured that FRRM was fit using the same training splits, sensitive variable encoding, and preprocessing steps applied to FERM. No post-hoc adjustments were made to FRRM outputs.

## F.2    ENVELOPE-BASED DECOMPOSITION (ALGORITHM 1)

Algorithm 1 requires estimation of two envelope subspaces: a response envelope of $X$ relative to $S$, and a predictor envelope of $Y$ relative to $X$.

**Envelope estimation:**    We estimate envelope bases using the objective function in (15), solved either with the 1D algorithm of Cook & Zhang (2016) or the NIECE algorithm of Zhang et al. (2023). In both cases, the estimated envelope subspace is $\sqrt{n}$-consistent. Dimensions $\hat{m}$ and $\hat{u}$ are chosen by BIC or cross-validation.

**Sample covariance estimators:**    Let $\hat{\Sigma}_Y$, $\hat{\Sigma}_X$, $\hat{\Sigma}_{XY}$, and $\hat{\Sigma}_{YX}$ denote the usual sample covariances. Then:

- *Response reduction:* $\hat{M} = \hat{\Sigma}_X, \hat{U} = \hat{\Sigma}_{XS} \hat{\Sigma}_{SX}$.
- *Predictor reduction:* $\hat{M} = \hat{\Sigma}_X, \hat{U} = \hat{\Sigma}_{XY} \hat{\Sigma}_{YX}$.

The intersections of the estimated envelope subspaces yield the four orthogonal components $\hat{\Pi}_{XSY}, \hat{\Pi}_{XY}, \hat{\Pi}_{XS}, \hat{\Pi}_0$.

**Numerical optimization:**    The envelope objective is non-convex and solved on Stiefel/Grassmann manifolds. In practice, we use manifold optimization routines from the R package `Renvlp`.

## F.3    INTERPOLATED REGRESSOR WITH FAIRNESS CONTROL (ALGORITHM 2)

Given the decomposition from Algorithm 1, Algorithm 2 trains regressors under three cases:

1. **Fair model** ($r = 0$): Only the subspace spanned by $\hat{\Pi}_{XY}$ is used, yielding predictions independent of $S$ asymptotically.

2. **Unconstrained model** ($r = 1$): OLS on projections of predictor $X$ on both subspaces spanned by $\hat{\Pi}_{XY}$ and $\hat{\Pi}_{XSY}$ subspaces. When $r = 1$, the statistically efficient choice remains to fit the model using only the material components ($X_{XY}$ and $X_{XSY}$), rather than the full predictor $X$. Even though these subspaces are estimated and therefore subject to sampling variability, prior empirical evidence Cook et al. (2010); Cook & Zhang (2015; 2016) and our experiments indicate that removing $Y$-immaterial variation typically provides efficiency gains that outweigh the additional estimation noise, particularly in moderate or high-dimensional settings.

3. **Interpolated model** ($0 < r < 1$): A ridge penalty is applied only to the shared component obtained by projecting $X$ on subspace spanned by $\hat{\Pi}_{XSY}$. The penalty $\lambda$ is chosen such that the $R^2$-fairness criterion in (8) satisfies $R_S^2(\lambda) \leq r$. We compute $\mathrm{Var}(\cdot)$ in (8) as the sample variance of centered fitted values.

**Practical computation of** $\lambda$**:** In practice, $\lambda$ is found via a line search (e.g., bisection) over a grid of candidate values until the fairness constraint $R_S^2(\lambda) \leq r$ is met. This ensures the fitted model interpolates smoothly between the fair and unconstrained extremes.

**Software:** All algorithms are implemented in R. For cross-validation and BIC, we rely on existing libraries for model selection. Ridge penalties are implemented using standard linear algebra solvers.

Together, Algorithms 1 and 2 provide a practical pipeline for training regressors that interpolate between fairness and predictive utility in a principled way.

**Additional advantage of our framework:** Unlike prior methods, FERM also decomposes the sensitive attributes $S$. This removes the immaterial variation in $S$ that is unrelated to $Y$, ensuring that fairness constraints target only the predictive overlap between $S$ and $Y$. As a result, our $R_S^2$ measure in (equation 8) provides a more principled notion of unfairness: it penalizes only the variance in predictions attributable to the material component of $S$, rather than noise. This refinement yields tighter fairness control and a sharper fairness-accuracy trade-off than existing approaches.

## G   ADDITIONAL DISCUSSION OF RELATED WORK

For completeness, we briefly situate FERM relative to two areas of the fairness literature that also discuss direct and indirect effects.

**Causal literature.** Works in the causal literature defines "direct" and "indirect" effects in terms of causal pathways under a structural causal model (e.g., Chiappa (2019); Pan et al. (2021)). These approaches rely on strong assumptions (e.g., valid mediators, no hidden confounding, identifiable counterfactuals) that are not available in our supervised learning setting with multivariate continuous $S$. Our decomposition is geometric - based on orthogonal subspaces of the $X$-space - rather than causal.

**Fair representation learning.** Methods from the fair representation–learning literature (e.g., adversarial learning, VFAE, pre-processing decorrelation) are almost exclusively developed for binary or categorical sensitive attributes and rely on iterative optimization procedures (e.g., Creager et al. (2019); Zemel et al. (2013); Liu et al. (2022)). Their loss functions and fairness objectives (e.g., adversarial equality of odds, demographic parity constraints, mutual-information penalties) are tailored to discrete protected groups and are not readily extendable to the multivariate continuous $S$ that we consider. Moreover, these approaches do not yield explicit fairness–utility trade-offs or closed-form decompositions, making them unsuitable as baselines for our theoretical framework.

## H   LIMITATIONS

While FERM provides a principled framework for fairness-aware regression, several limitations should be noted:

**Linearity of subspace decomposition:** FERM assumes that both material and immaterial variation can be captured through *linear* subspaces of the predictor space. In many real-world applications, sensitive attributes may influence $Y$ through nonlinear interactions with $X$, in which case a purely linear envelope decomposition may be too restrictive. Extending the methodology to nonlinear settings (e.g., kernelized or deep envelope methods) remains an important direction for future work.

**Dependence on subspace estimation:** The validity of FERM relies on accurately estimating the envelope subspaces. Although the 1D and NIECE algorithms provide $\sqrt{n}$-consistent estimators, in finite samples the estimated subspaces may deviate from their population counterparts, especially when $d_X$ is large relative to $n$. This may lead to imperfect fairness guarantees in small samples.

**Choice of tuning parameters:** FERM requires selecting the envelope dimensions $(\hat{m}, \hat{u})$ and the interpolation parameter $\lambda$ (or equivalently the fairness target $r$). While BIC and cross-validation are standard, their stability can vary across datasets. Automatic and robust criteria for selecting these parameters remain an open area of research.

**Fairness notion:** Our fairness control is expressed through covariance-based independence (or $R_S^2$) between predictions and sensitive attributes. Other fairness notions (e.g., counterfactual fairness, equalized odds) are not directly captured. Extending the envelope framework to these broader definitions requires additional methodological development.

**Computational cost:** Envelope estimation involves non-convex optimization on Stiefel or Grassmann manifolds. Although efficient algorithms like NIECE exist, the procedure is more computationally intensive than standard regression, and scalability to very high-dimensional predictors may be challenging without additional structural assumptions.

Overall, while these limitations highlight directions for future work, FERM demonstrates that subspace decomposition provides a tractable and interpretable pathway to fairness-aware regression.

## LLM USAGE

Large Language Models (LLMs) were used as an assistive tool in preparing this submission. Their role was limited to: (i) editing and polishing drafts of the introduction and methodology sections for clarity and conciseness; (ii) suggesting alternative phrasings to improve readability; and (iii) checking grammar, spelling, and typographical errors. All technical contributions, theoretical results, experimental designs, and analyses were developed entirely by the authors. The authors take full responsibility for the accuracy and integrity of the content.

