# OpenReview forum: "Achieving Fairness-Utility Trade-offs through Decoupling Direct and Indirect Bias"
_ICLR.cc/2026/Conference — Submitted to ICLR 2026_

### Official Review · Reviewer_w8Fb · 2025-10-29

**Soundness:** 3
**Presentation:** 3
**Contribution:** 2
**Rating:** 4
**Confidence:** 3

**Summary:**

This paper proposes a method to improve fairness in regression through decomposing the input space into components that vary with the label Y and/or a sensitive attribute S. Theoretically they show how this can be estimated consistently and show it is a variance-reducing projection. Empirically, they demonstrate some wins on simulated and real data in some combination of predictive MSE and fairness.

**Strengths:**

- believe there is novelty in the algorithm, in the application of envelope regression to fair regression, and seems like a good fit of techniques
- synthetic experiment is compelling, showing a nice win in predictive MSE that one doesn't usually see in fairness papers
- theoretical results seems sound, and the interpolation/decomposition result in 5.4 seems intuitive and useful

**Weaknesses:**

- framing: authors introduce the idea of this paper as decomposing direct/indirect bias and I'm not quite sure that the method actually touches on that. For instance, indirect bias could be present in either XS or XSY as either can contain proxy variables.
- related work: it would be nice to see more discussion of + comparison to a) work from the causal literature that claims to do direct/indirect bias decomposition (are these the same/different ideas?) b) work from the fair representation learning literature that aims to learn de-correlated predictors through pre-processing
- some lack of clarity around (1) and (2) in the Limitations section in L185 - I don't think the rest of the paper really demonstrates how we get efficiency or interpretability gains with this method; certainly there isn't an empirical demonstration
- background: it would be great to get more background on envelope regression in the main body, given that it's the central technical tool in this paper. What does it do, and how should I think about it?
- Alg 2: not clear to me why if we have r=1, we wouldn't be fitting just normal OLS on all of X (given that the XS-estimation process is probably noisy, doing the unconstrained thing may be better)
- in Sec 6, should give more clarification on the difference in the two FERM methods: is one on XY + XSY and the other on XY only? something else?
- In general in the experiments, would be good to have more comments on what I should be looking for here - better MSE? better fairness? both? for instance in the Fig 3, we don't see an MSE improvement, or much of an unfairness improvement - would be helpful to communicate better what in the graph I should be taking away


smaller points:
- Fig 1 is more confusing than helpful I think - I'd recommend visualizing X as a vector rather than a space
- “re Γ spans directions of X associated with S and Γ0 its invariant complement; Φ spans predictive directions for Y and Φ0 the immaterial ones.” - not sure why the authors use different terminology for Y and S? Are the concepts different? (eg associated/predictive, invariant/immaterial)
- L313: would be good to get more clarity on what it means to "exploit the envelope structure",  as well as the difference between the approaches outlines in equations on L313 and L315
-

**Questions:**

what is the exact relationship of the method  to direct/indirect bias decomposition?
how does the method improve on the 2 limitations from L185?
what does envelope regression do, and how?
what should the reader be looking for in the experiments?

---

> ### Author Response · Authors · 2025-11-21
> **Rebuttal 1**
>
> We sincerely thank the reviewer for their thoughtful and constructive comments, as well as for recognizing the novelty of our FERM framework to fair regression, the strength of the theoretical results, and the compelling nature of the synthetic experiments. We address the main concerns below.
>
> ---
>
> **Response to Weakness W1, W2, W3**
>
> ---
>
> **(W1) Framing:** You are correct under the usual definitions: in many existing works, components like the $\Pi_{X S}$  and $\Pi_{X SY}$ - subspaces are both counted as sources of indirect bias, because they are statistically linked to the sensitive attribute $S$.
>
> However, in our framework the $\Pi_{X S}$-subspace is by construction non-predictive for $Y$: it carries information about $S$, but no additional information about $Y$. Therefore, including it in the prediction would only inflate variance without improving prediction accuracy. That is why we explicitly drop the $\Pi_{X S}$ subspace when modeling $Y$, which is the key novelty of our method.
>
> That is, relative to standard "direct/indirect bias" decompositions, we refine these notions by looking at the intersection of (direct/indirect) sensitive components with the part of the feature space that is actually predictive of $Y$. Only those $Y$-informative sensitive components matter for the fairness-utility trade-off, leading to guaranteed statistical efficiency gain, as demonstrated in Theorem 5.1.
>
>
> ---
>
> **(W2) Related work:** We appreciate the reviewer’s suggestion to relate our work to (a) causal approaches that decompose direct and indirect effects and (b) fair representation-learning methods aimed at de-correlating predictors. These lines of research are indeed conceptually related, but they are not directly applicable to our setting for the following reasons.
>
> First, work in the causal literature defines “direct” and “indirect” effects in terms of causal pathways under a structural causal model. These approaches rely on strong assumptions (e.g., valid mediators, no hidden confounding, identifiable counterfactuals) that are not available in our supervised learning setting with multivariate continuous $S$. Our decomposition is geometric - based on orthogonal subspaces of the $X$ -space - rather than causal.
>
> Second, methods from the fair representation–learning literature (e.g., adversarial learning, VFAE, pre-processing decorrelation) are almost exclusively developed for binary or categorical sensitive attributes and rely on iterative optimization procedures. Their loss functions and fairness objectives (e.g., adversarial equality of odds, demographic parity constraints, mutual-information penalties) are tailored to discrete protected groups and are not readily extendable to the multivariate continuous $S$ that we consider.
> Moreover, these approaches do not yield explicit fairness–utility trade-offs or closed-form decompositions, making them unsuitable as baselines for our theoretical framework.
>
> We have added a brief discussion clarifying these distinctions in the Appendix, Section G of the revised paper.
>
> ---
>
> **(W3) Efficiency and interpretability:**
>
> We appreciate the reviewer’s feedback regarding (1) efficiency and (2) interpretability in the Limitations section (L185). Indeed, our framework is designed precisely to address these two limitations of existing methods, which we summarize here:
>
> **Efficiency.** Theorem 5.1 formally establishes the efficiency gains of our approach: whenever an envelope structure is present, the envelope estimator achieves **strictly smaller asymptotic variance** than OLS by excluding immaterial variation in $X$. This is fully aligned with the classical regression principle that removing irrelevant or noise-like components can improve estimator efficiency and, in turn, lead to better predictive performance. The predictive MSE patterns we observe in our experiments (see Figures 2 and 3) are consistent with this variance reduction: FERM frequently exhibits smaller MSE than FRRM across sample sizes, which is exactly what one expects when coefficient estimators have lower asymptotic dispersion.
>
> **Interpretability.** In contrast to black-box, optimization-based fair representation–learning algorithms, our method explicitly decomposes the $X$-space into four orthogonal (and hence non-overlapping) subspaces: shared, predictive, sensitive, and residual. When constructing the prediction model for $Y$, we deliberately use only the predictive subspace ($\Pi_{XY}$), together with a controlled portion of the shared subspace ($\Pi_{XYS}$) to achieve intermediate fairness levels. This yields an inherently interpretable structure: one can see precisely which components of $X$ drive prediction, which carry sensitive information, and how the trade-off between fairness and utility is implemented in terms of these subspaces.
>
> We have revised the methodology and theoretical properties sections to make these efficiency (Line 326-328) and interpretability (Line 254-259) advantages more explicit.

---

> ### Author Response · Authors · 2025-11-21
> **Rebuttal 2**
>
> **Response to Weakness W4-W7**
>
> ---
>
> **(W4) Envelope background:** Due to space constraints, we have deferred technical details of envelope estimation in Appendix A.
> In the revision, we will add a concise summary in Section 4.1 explaining that an envelope identifies the **minimal subspaces** of $X$ that is material for predicting $Y$ and correlated with $S$ (Cook and Zhang, 2015; 2016). FERM constructs two envelopes - one $S$-associated and one $Y$-predictive, and uses their intersections and complements to obtain the four-way decomposition. Technical details remain in Appendix A. In the revised version, we have **added Figure 4** in Appendix A to **illustrate the idea of envelopes**.
>
> ---
>
> **(W5)** In Algorithm 2, when $r = 1$, we are not fitting an OLS model on the full feature vector $X$. Instead, as defined in Lines 290-291, $r = 1$ corresponds to using only the useful components of $X$ for predicting $Y$, namely the
> $X_{XY}$ and $X_{XSY}$ subspaces (Eq. 6). These are the components that carry information about $Y$ after removing the purely sensitive and non-predictive parts of the $X$ space.
>
> In other words, fitting an unconstrained OLS model on all of $X$ is typically suboptimal (as we establish in Theorem 5.1) and this is further supported by our simulations: removing prediction-irrelevant variation in the $X$-space leads to substantial gains in predictive accuracy, especially in moderate to high dimensions, even when the subspaces are estimated. This behavior is consistent
> with prior empirical findings in the sufficient dimension reduction literature (e.g., Cook and Zhang, 2015; 2016), which show that the gains in efficiency and predictive performance from focusing on $Y$-relevant subspaces outweigh the additional noise introduced by estimation.
>
> We have added this clarification to Appendix E.3 (case $r = 1$), where we also provide additional implementation details.
>
> ---
>
> **(W6)** As clarified in Lines 363–366, the two FERM variants differ in which subspaces of $X$ are retained for prediction:
>
> **FERM–decorrelated** removes only the $X_{XS}$ component, i.e., it uses $X - X_{XS}$, which contains both the predictive and residual components.
>
> **FERM–predictive** removes both $X_{XS}$ and $X_{0}$, retaining only the predictive directions. In other words, FERM–decorrelated uses the (predictive + residual) subspace, whereas FERM–predictive uses the predictive subspace only.
>
> As demonstrated in our experiments, FERM–predictive achieves substantially lower MSE in commonly encountered moderate to high dimensional setting, highlighting the benefit of our finer $X$-space decomposition and the removal of prediction-irrelevant components.
>
> ---
>
> **(W7)** Thank you for pointing this out. The apparent lack of improvement in Figure 3 was due to a captioning error in the submitted version: the captions for the two subplots were inadvertently swapped. We have corrected this in the revision.
> With the corrected captions, FERM consistently outperforms FRRM in predictive MSE across all sample sizes and fairness thresholds.
>
> To clarify what the reader should focus on in Figures 2 and 3 (and we have expanded the discussion in Lines 424–451 to make these points clearer):
>
> **Fairness–utility trade-off (left panels).** At any fixed unfairness budget $r$, both variants of FERM achieve lower test MSE
> than FRRM. The gain is especially pronounced for FERM–predictive in higher-dimensional settings, reflecting the benefit of removing prediction-irrelevant components.
>
> **Attained unfairness (right panels).** All methods track the target unfairness budget $r$ closely on the test set,
> showing that the fairness constraint is accurately enforced.
>
> ---
>
> **References**
>
> 1. R Dennis Cook and Xin Zhang. Foundations for envelope models and methods. Journal of the American Statistical Association, 110(510):599–611, 2015.
>
> 2. R Dennis Cook and Xin Zhang. Algorithms for envelope estimation. Journal of Computational and Graphical Statistics, 25(1):284–300, 2016.
>
> ---

---

> ### Author Response · Authors · 2025-11-21
> **Rebuttal 3**
>
> **Response to smaller points**
>
> ---
>
> 1. Thank you for this suggestion. We agree that visualizing $X$ as a vector can be useful in some contexts; however, in our setting the central object of interest is the *space* spanned by the columns of $X$, not an individual realization of $X$. Our methodology and all theoretical guarantees are formulated at the level of subspaces: the shared, predictive, sensitive, and residual components are defined as orthogonal *subspaces* of the $X$-space. For this reason, Figure 1 illustrates a decomposition of the predictor space rather than a single vector, as the geometry of these subspaces is fundamental to understanding our approach.
>
> To improve clarity, we have **revised Figure 1** and the accompanying caption to emphasize that the decomposition applies to the $X$-space as a whole and not to a specific observation. We believe this more accurately reflects the concepts of the proposed method.
>
> ---
>
> 2. We use the terms "associated'' for $S$ and ``predictive'' for $Y$ because the two envelopes play fundamentally different roles in our framework. The matrix $\Gamma$ spans directions of $X$ that are *associated* with $S$, that is, the
> variation in $X$ that can be explained by $S$. In contrast, $\Phi$ spans the directions of $X$ that are *predictive* of $Y$, i.e., the variation that is material for the regression of $Y$ on $X$.
>
> Their complements naturally take on different names: $\Gamma_0$ corresponds to $S$-invariant directions, while $\Phi_0$ corresponds to $Y$-immaterial directions. This terminology is directly inherited from the envelope regression literature, where predictor envelopes (for $S$) and response envelopes (for $Y$) describe these two distinct types of material–immaterial decompositions.
>
> We agree that this distinction can be stated more explicitly, and we have clarified the terminology in the revision (Remark A.2) to make the relationship between the two envelope types more clear.
>
> ---
>
> 3. Thank you for the helpful question. We clarify here in more details what it means to ``exploit the envelope structure'' and the distinction between the approaches described in Lines 313 and 315. The expression in Line 313 corresponds to residual-based approaches that remove only the $S$-associated component $X_{XS}$ and then regress on the remaining part, $X - X_{XS} = X_{XSY} + X_{XY} + X_0.$ This approach discards the sensitive-only component but does not differentiate among the predictive directions ($X_{XY}$), the predictive–sensitive directions ($X_{XSY}$), and the immaterial noise ($X_0$). Consequently, OLS is still fitted using directions that contain no useful information for predicting $Y$, which may reduce stability and efficiency.
>
> **“Exploiting the envelope structure’’** (Line 315) refers to using the predictor and response envelopes to explicitly isolate the *material* subspace for predicting $Y$. Our method regresses only on $X_{XSY} + X_{XY}$, removing both the sensitive-only component and the immaterial variation $X_0$. This targeted use of the envelope structure is precisely what enables improved estimation efficiency and stability, as formalized in Theorem 5.1.
>
> We will revise the text around Lines 313–315 to make this distinction and the role of the envelope structure more explicit.
>
> ---

---

> ### Author Response · Authors · 2025-11-21
> **Rebuttal 4**
>
> ---
>
> **Response to (Q1)**
>
> ---
>
> As detailed in our responses to W1, W3, W4, and W7 above, we summarize the key points here.
>
> **Direct vs.\ indirect bias.** In the observational fairness sense used in Lines 34–37, direct bias refers to dependence on $S$ itself, and indirect bias refers to dependence on predictors correlated with $S$. Our decomposition makes these contributions explicit: $X_{XS}$ captures $S$-aligned but $Y$-immaterial variation (indirect bias that only adds noise), $X_{XSY}$ captures predictive-sensitive variation (indirect bias that affects $Y$), and $X_{XY}$ captures predictive variation that is independent of $S$. This separation is not available in prior fair-regression methods, which treat all $S$-correlated directions uniformly.
>
> **How FERM addresses the L185 limitations.** The inefficiency noted in L185 arises from FRRM’s residual-based decomposition, which ignores predictor correlations and retains immaterial variation. FERM overcomes this by using envelope regression to remove the $Y$-immaterial components ($X_{XS}$ and $X_{0}$) in an optimal way, yielding lower asymptotic variance (Theorem~5.1). The interpretability issue in L185 also disappears: the four orthogonal components ($\Pi_{XY}$, $\Pi_{XS}$, $\Pi_{XSY}$, $\Pi_0$) explicitly show how sensitive and predictive information are organized in $X$.
>
>  **What envelope regression does.** An envelope identifies the *minimal subspace of $X$ that is material for predicting $Y$*; directions orthogonal to this subspace contribute only noise (Cook and Zhang, 2015; 2016). FERM constructs two such envelopes - one associated with $S$ and one predictive of $Y$ - and their intersections and complements yield the four-way decomposition above.
>
> **What to look for in the experiments.** The left panels of Figures 2 - 3 show the fairness–utility trade-off: at any fixed unfairness budget $r$, FERM achieves lower or comparable MSE than FRRM, with FERM–predictive outperforming FERM–decorrelated in higher dimensions. The right panels show that the realized unfairness closely matches the target $r$. Together, these results illustrate exactly the benefits predicted by the theory: improved efficiency through removal of immaterial variation and precise control of sensitive directions through the shared predictive–sensitive component.
>
> ---
>
> We hope that these clarifications address all of the reviewer’s comments. We are grateful for the thoughtful feedback, which has helped improve the clarity and presentation of the paper.
>
> ---

---

### Official Review · Reviewer_wL7E · 2025-11-01

**Soundness:** 4
**Presentation:** 4
**Contribution:** 3
**Rating:** 8
**Confidence:** 3

**Summary:**

This paper presents a framework for using subspace decomposition using envelope regression to improve the fairness in regression algorithms. By decomposing a predictor space into four components and isolating the part that corresponds to sensitive attributes, this provides a way to distinguish direct and indirect bias. The model introduces a ridge penalty to the sensitive components, which theoretical and empirical results show a way to trade-off between accuracy and fairness that improves on prior methods.

**Strengths:**

S1) The paper presents a novel and principled decomposition to address an important problem.

S2) Theoretical results are strong and show smaller asymptotic variance.

S3) Empirical results are clear and compelling

**Weaknesses:**

W1) The framework relies on linear subscape decompositions. Further discussion about this assumption and the prevalence in real-world settings would be helpful.

**Questions:**

Other clarification questions:
Q1) I can't find additional implementation details about the baselines including hyperparameter tuning, and that would be helpful for understanding the experiments better.

---

> ### Author Response · Authors · 2025-11-20
> **Rebuttal 1**
>
> We sincerely thank the reviewer for their positive and encouraging assessment of our work. We appreciate the recognition of the novelty of our decomposition, the strength of our theoretical guarantees, and the clarity of the empirical results. We also thank the reviewer for providing thoughtful suggestions that help us further clarify key aspects of our method.
>
> ---
>
> **(W1) Why linear subspace decomposition?**
> As you also note in your other comments, our framework decomposes the $X$ -space into four orthogonal (and hence non-overlapping) subspaces: shared, predictive, sensitive, and residual, which is the first in the literature. Obtaining such an **explicit, closed-form decomposition**, together with a **closed-form fairness–utility trade-off**, **requires a linear model** for $Y$ as a starting point. This is directly analogous to how PCA is developed: classical PCA begins with a linear subspace decomposition, and only after the linear theory is fully established do extensions such as kernel PCA address nonlinear structure. The linear theory provides clarity, interpretability, and closed-form solutions, which then serve as the foundation for more flexible nonlinear variants. Our contribution plays an analogous role: it establishes the linear case as a principled and transparent baseline with theoretical guarantees.
>
> Of course, an **extension to nonlinear relationships** (e.g., via kernel methods) is possible. Our Appendix C already outlines several routes for extensions, including kernelized envelopes and neural parameterizations of the envelope mapping - that preserve the subspace-separation principle. We have **expanded this discussion** in the revised version in the **Appendix C** clarifying when linear envelopes are appropriate and how these nonlinear generalizations can be applied. Overall, we view the linear formulation as a principled and interpretable first step on which these extensions can be built. Finally, we want to point out if one is not concerned with preserving the closed-form fairness-utility trade-off we establish, one could instead use a neural network for modeling $Y$ based on the resulting components from our $X$-space decomposition to improve prediction accuracy.
>
> At the same time, linear subspace structures are widely encountered in real-world tabular prediction tasks after standard preprocessing steps such as centering, encoding, and basic dimension reduction. A major practical advantage of envelope methods - especially in **high-dimensional applications**- is their goal of identifying a low-dimensional linear subspace that captures all material variation for prediction. Prior work in envelope regression (Cook et al., 2010; Cook \& Zhang, 2016; Zhang et al., 2022) shows that the dependence between covariates and the response is often well captured by **low-dimensional linear components**. In fairness applications, this has an additional benefit: it yields a transparent, **auditable separation between direct and indirect bias**, which is crucial in domains where **interpretability** and regulatory accountability matter.
>
> ---
>
>
> **(Q1)**
> In the revision, we have added full implementation details of the baseline Fair Ridge Regression Model (FRRM) in Appendix F.1. For FRRM, we use the authors’ implementation (`frrm` function in `library(fairml)` in `R`) exactly as recommended. Specifically, FRRM takes the desired unfairness level $r$ as input, and the algorithm internally selects the corresponding ridge penalty $\lambda$. In our experiments (Section 6), we used $r = \{0,0.1,0.2,0.3,0.4,0.5\}$. No further hyperparameter tuning is required.
>
> ---
>
> We appreciate the reviewer’s supportive evaluation and insightful suggestions. We hope that the additional clarification and expansions in the revised manuscript address all concerns.
>
> ---
>
> **References:**
>
> 1. R Dennis Cook and Xin Zhang. Foundations for envelope models and methods. Journal of the American Statistical Association, 110(510):599–611, 2015
>
> 2. R Dennis Cook and Xin Zhang. Algorithms for envelope estimation. Journal of Computational and
> Graphical Statistics, 25(1):284–300, 2016.
>
> 3. Xin Zhang, Kai Deng, and Qing Mai. Envelopes and principal component regression. Electronic Journal of Statistics, 17(2):2447–2484, 2023.
>
> ---

---

### Official Review · Reviewer_eWet · 2025-11-01

**Soundness:** 3
**Presentation:** 2
**Contribution:** 2
**Rating:** 4
**Confidence:** 5

**Summary:**

In this paper, the authors consider fairness in regression and state the importance of fairness. They introduce a fairness framework that adapts subspace decomposition techniques from envelope regression. The predictor space is decomposed into four orthogonal components: response-specific variation, sensitive variation, shared variation, and residual noise. This angle is good. Moreover, this decomposition makes it more interpretable in fairness regression.

**Strengths:**

- The decomposition is good, which makes it more interpretable in fairness learning.
- This paper provides numerical experiments on simulated and real datasets.

**Weaknesses:**

- Only consider the linear relationship between response and feature $X$.
- The authors should give the full names when they use at the first time.

**Questions:**

- Only consider the linear relationship between response and feature $X$. My main concern is about non-linearity. When we evaluate on tubular datasets (real datasets), 3-layer or 4-layer fully connected neural networks are used. In this paper, the author decompose the predictor space into 4 parts and consider linear regression on 2 of them.
- ``Envelope'', what is the meaning of this word?
- Lines 318-319, is it a definition of asymptotic variance matrix? Also, $T$ is a random variance? $\theta$ is a mean value? I am not sure my guess is right or wrong?
- Line 161, there is a mistake about $Cov(S,\hat U)=0$, it should be $S\perp\hat U$, since you need to subtract the mean values when calculating covariance?

---

> ### Author Response · Authors · 2025-11-20
> **Rebuttal 1**
>
> We sincerely thank the reviewer for their constructive feedback and for highlighting the interpretability benefits of our structured decomposition. We appreciate the comments and the opportunity to explain our modeling choices and notation in more detail.
>
> ---
>
> **(W1) Linear relationship and nonlinearity:**
> As you also note in your other comments, our framework decomposes the $X$-space into four orthogonal (and hence non-overlapping) subspaces: shared, predictive, sensitive, and residual, which is the first in the literature. Obtaining such an explicit, **closed-form decomposition**, together with a **closed-form fairness–utility trade-off**, **requires a linear model** for $Y$ as a starting point.
>
> This is directly analogous to how PCA is developed: classical PCA begins with a linear subspace decomposition, and only after the linear theory is fully established do extensions such as kernel PCA address nonlinear structure. The linear theory provides clarity, interpretability, and closed-form solutions, which then serve as the foundation for more flexible nonlinear variants. Our contribution plays an analogous role: it establishes the linear case as a principled and transparent baseline with theoretical guarantees.
>
> Of course, an **extension to nonlinear relationships** (e.g., via kernel methods) is possible. Our Appendix C already outlines several routes for extensions, including kernelized envelopes and neural parameterizations of the envelope mapping - that preserve the subspace-separation principle. We have expanded this discussion in the revised version in the Appendix C clarifying when linear envelopes are appropriate and how these nonlinear generalizations can be applied. Overall, we view the linear formulation as a principled and interpretable first step on which these extensions can be built.
>
> Finally, we want to point out if one is not concerned with preserving the closed-form fairness–utility trade-off we establish, one could instead use a neural network for modeling $Y$ based on the resulting components from our $X$-space decomposition to improve prediction accuracy (Appendix C).
>
> ---
>
> **(W2)** We have revised the entire manuscript to ensure that all acronyms are introduced with their full names at first use.
>
> ---
>
> **(Q1)** See our reply to your earlier comments in Weakness 1.
>
> FERM uses only the two components of $X$ that are predictive for $Y$ ($X_{XY}$ and $X_{XSY}$), while removing the other two components ($X_{XS}$ and $X_{0}$) that carry prediction-irrelevant information in X space. This **selective use of only the predictive-relevant subspaces** is precisely what leads to the **variance reduction** reflected in Theorem 5.1 and in our empirical results.
>
> ---
>
> **(Q2) Envelope:** The formal definition of an envelope is provided in Appendix A (Definition A.1). In our context, an “envelope’’ is a term from envelope regression
> (Cook et al., 2010) that refers to the smallest subspace of the predictor (or response) space that
> contains all variation relevant for a given regression task. Intuitively, an envelope extracts the
> directions that matter for predicting $Y$ or capturing dependence on $S$, while
> discarding prediction-irrelevant information in X space. In the revised version, we have added Figure 4 to illustrate this idea of envelopes.
>
> ---
>
> **(Q3) Asymptotic variance:**
> Lines 318–319 introduce the definition of the asymptotic variance matrix. Here, $T$ denotes a generic estimator (e.g., $\hat{\beta}_{OLS}$), and $\theta$ denotes the corresponding population parameter (e.g., the true regression coefficient $\beta$).
>
> $T$ is not a random variance; rather, the expression describes the asymptotic distribution of the estimator $T$ around its target $\theta$, whose covariance structure is given by the asymptotic variance matrix.
>
> ---
>
> **(Q4) Covariance and centering:**
> Our statement is correct under our setting, where both $X$ and $S$ are centered.
> Recall, in Equation (1), $X = SB + U$, the OLS residuals $\hat{U}$ satisfy $S^\top \hat{U} = 0$ by construction.  When $S$ and $X$ are centered, this orthogonality directly implies $\operatorname{Cov}(S,\hat{U}) = 0$.
> To avoid confusion, we have added that $X,S$ and $Y$ are centered (zero mean) earlier in Section 3 in the revised paper.
>
> ---
>
> We hope that these clarifications address all of the reviewer’s comments. We are grateful for the thoughtful feedback, which has helped improve the clarity and presentation of the paper.
>
> ---

---

### Official Review · Reviewer_WTmm · 2025-11-01

**Soundness:** 3
**Presentation:** 2
**Contribution:** 3
**Rating:** 4
**Confidence:** 4

**Summary:**

The paper proposes FERM, a fairness-aware regression method which decomposes the predictor space into response-specific, sensitive, shared and residual variations using envelope regression, in order to disentangle direct and indirect biases. FERM applies Ridge penalty only to the sensitive subspace to yield fine-grained, interpretable control over how sensitive attributes influence prediction. Authors provide theoretical results about consistency and efficiency of estimation with provable reductions in asymptotic variance relative to OLS and provide a closed-form characterization of fairness and utility. While the theory is sound, empirical validations are limited as they provide only one baseline, which is not consistently inferior to proposed method (example for smaller sample size or larger threshold $r$). I believe the paper could benefit from more extensive experiments to (i) validate theoretical claims about consistency and nonlinear robustness, (2) consider additional baselines and real datasets, and (iii) identify conditions under which the proposed method is superior to existing ones. The presentation of the paper could improve.

**Strengths:**

- Envelope estimation enables more statistically efficient (lower-variance) estimators and grounded understanding of model components.

- Applying a ridge penalty only to the shared (sensitive + response) subspace permits continuous interpolation between full fairness (no sensitive influence) and unconstrained prediction (maximum utility); assign Eq (6).

- The proposed decomposition is structured for a multivariate setting, which permits handling multiple sensitive attributes, which could generalize better than pairwise or moment constraints.

- Theoretical guarantees of efficiency and consistency, as well as closed-form for fairness-accuracy tradeoff, are provided in Section 5.

**Weaknesses:**

- Empirical validation is very limited--testing on only one real dataset and comparing against a single baseline--leaving method's generalization questionable and limiting strength of conclusions. Including a broader range of conceptually aligned fair regression methods and more real datasets will improve validation.

- In the real-data experiment, FERM and the baseline FRRM exhibit "somewhat" comparable performance, with FRRM occasionally outperforming FERM at some unfairness thresholds (levels). This suggests that FERM’s practical benefits may diminish outside controlled settings. Further validation is needed.

- While the experiments validate improved efficiency and some fairness–accuracy trade-off, they fall short of evaluating other theoretical claims such as estimator convergence and asymptotic fairness as established in Proposition 5.2 and Lemma 5.3.

- The synthetic experiment design at beginning of Sec 6 is confusing (not aligned with regression models in Eqns (4)-(6)). Both FERM-decorrelated and FERM-predictive sound different than the interpolated model, yet both apply a Ridge penalty which according to theory, should be on shared subspace. Also, both perform somewhat similarly.

- Wouldn't the controlled linear dependence in data generation imply separable predictive & sensitive subspaces, hence the penalty could be unnecessary. Evaluating results without the penalty may clarify whether fairness arises from the decomposition itself or from regularization.

**Questions:**

Please see weaknesses section.

---

> ### Author Response · Authors · 2025-11-20
> **Rebuttal 1**
>
> We sincerely thank the reviewer for their rigorous critique and detailed feedback. We value their recognition of our methodological contributions, including the structured envelope-based decomposition, the interpretability afforded by penalizing only the sensitive subspace, and the accompanying theoretical guarantees. We also appreciate the opportunity to clarify and strengthen the points the reviewer raised regarding our empirical evaluations.
>
> ---
>
> **Response to Weakness W1, W2, W3:**
>
> ---
>
> **(W1)** **Comparison with only one baseline:** As discussed in Sections 2 and 3.2, there are no other conceptually aligned baselines beyond FRRM (Scutari et al., 2022), which has already been shown to outperform earlier frameworks such as Komiyama et al. (2018). This makes FRRM the uniquely appropriate comparator in our context.
>
> **One real dataset** Given the theoretical nature of our contribution, i.e., a principled feature-space decomposition framework with explicit fairness–utility tradeoff guarantees, our empirical studies follow a standard practice for theory-driven work: we combine a representative real-data study with extensive and systematically varied synthetic experiments to validate the main insights. With that said, we agree that additional real datasets would further strengthen the empirical section, and we have **incorporated additional real-data experiment** in the revision (see Appendix E.2). (New real data experiment: Full-Year Consolidated (FYC) datafiles available at Medical Expenditure Panel Survey (MEPS) [https://meps.ahrq.gov/data_stats/download_data_files.jsp], which combine person-level demographics, socioeconomic status, insurance coverage, and all medical expenditures across the entire year.)
>
> ---
>
> **(W2) Interpretation of the real-data results:** We apologize for the confusion; the captions of the two subfigures in Figure 3 were inadvertently swapped and have been corrected in the revised paper.
>
> We agree that in some scenarios the two methods exhibit similar performance. As mentioned in Theorem 5.1 (Line 324), the efficiency gains of FERM are most pronounced when the overlaps among the $X$–$S$ and $X$–$Y$ spaces are minimal. That is, when these overlaps are substantial, FERM (ours) and FRRM are expected to behave similarly, and we have deliberately designed our simulation studies to reflect this behavior honestly (e.g. Figure 2 (1a), $r = 0.1$). On the other hand, when these overlaps are moderate (the setting to be expected in practice), we have shown that FERM consistently outperforms FRRM in predictive MSE across all sample sizes and fairness thresholds.
>
> ---
>
> **(W3) Empirical evaluation of Proposition 5.2 and Lemma 5.3:**  The results in Proposition 5.2 and Lemma 5.3 are included to provide theoretical completeness, but they are not the main focus of the paper, which focuses on establishing theoretical guarantees for the fairness–utility trade-off. This is why they are stated as propositions and lemmas rather than as central theorems.
>
>   However, we have followed your suggestion and **added** in Appendix D the plots:
>
> (i) **showing empirical consistency** of the envelope estimator $\hat{\beta}_{env}$ (Proposition 5.2).
>
> (ii) **convergence** of $Cov(\hat{Y},S) \to 0$ as $n$ increases (Lemma 5.3 (asymptotic fairness)).
>
> (iii) ratio $ \mathrm{Var}(\hat{\beta}{\mathrm{OLS}}) / \mathrm{Var}(\hat{\beta}{\mathrm{env}}) $, for varying sample sizes, unfairness levels $r$, and $d_X$ (ratio is greater than or equal to 1), demonstrating **consistent variance reduction** as shown in Theorem 5.1.
>
> ---

---

> ### Author Response · Authors · 2025-11-20
> **Rebuttal 2**
>
> **Response to Weakness W4 & W5:**
>
> ---
>
> **(W4) Synthetic experiment design:** As discussed in Lines 363–366: (i) FERM–decorrelated uses $X - X_{XS}$ (predictive + residual components), and (ii) FERM–predictive uses $X - X_{XS} - X_{0}$ (predictive only). By comparing these two variants, one can clearly examine the **benefit of removing residual subspace**, i.e., the irrelevant information in the prediction of $Y$. With these two choices, we then include a ridge penalty (Lines 367–368) on the sensitive components to achieve intermediate unfairness levels ($0 < r < 1$).
>
> As further discussed in Lines 427–429, the two variants behave similarly, as expected, in low-dimensional settings where the overlap structure is large (i.e., the residual subspace is small). In higher dimensions (**the common scenario in practice**), however, FERM–predictive achieves distinctly lower predictive MSE because the overlap is smaller (i.e., the residual subspace is larger), demonstrating the benefit of removing prediction-irrelevant information.
>
> ---
>
> **(W5) Overlap of predictive and sensitive subspaces:** The fact that $Y$ depends linearly on $X$ and $S$ does not imply that $X$ and $S$ are independent. In our data-generating process, $X$ and $S$ are correlated, so there is overlap between the $X$-space and the $S$-space. Recall that we decompose the entire space of $X$ into four orthogonal (and hence non-overlapping) subspaces: shared, predictive, sensitive, and residual. By construction, the predictive and sensitive subspaces themselves do not overlap, even though the original $X$ and $S$ spaces do. This decomposition is precisely what allows the ridge penalty on the sensitive-aligned directions to control the fairness–utility trade-off.
>
> ---
>
> We hope that our additional experiments and clarifications satisfactorily address all of the reviewer’s concerns, and we sincerely appreciate the thoughtful feedback.

---

### Author Response · Authors · 2025-12-02
**Author Summary of Revisions and Response to Reviewers 1**

**Summary of Contributions:**

To assist the Area Chair in their assessment, we briefly recap the core contributions of our work, which establishes a foundational statistical framework for fair regression:

1. **Novel 4-Way Decomposition:** We are the first to rigorously **decompose the feature space** into four **orthogonal** and **interpretable subspaces**: predictive, sensitive, shared, and residual. This geometric structure provides a transparent mechanism to distinguish between direct bias, indirect bias, and prediction-irrelevant noise, which prior approaches lack.

2. **Theoretical Efficiency:** By leveraging envelope regression, FERM provides provable **statistical guarantees**, achieving strictly smaller asymptotic variance than standard estimators (Theorem 5.1). This offers a principled alternative to heuristic regularization methods.

3. **Complex Fairness Settings:** We address the challenging setting of **multivariate, continuous sensitive attributes**, filling a significant gap in the literature where most methods are restricted to discrete or binary protected groups or univariate sensitive attributes.

4. **Foundational Linear Baseline:** We establish the rigorous theory for the linear case - providing closed-form solutions and interpretability - which serves as the necessary foundation for future work extending to non-linear settings (e.g., via Kernel methods), as outlined in our Appendix.

---

**Summary of Revisions:**

We thank the reviewers for their constructive feedback. We are encouraged that all reviewers recognized the **novelty of the FERM framework**, the **theoretical guarantees**, and the **interpretability** of isolating sensitive vs. predictive subspaces.
However, we believe the initial borderline ratings do not fully reflect these acknowledged contributions, particularly in light of our revisions. We have addressed every raised concern - adding new real dataset results and empirical validation of our theoretical results  - and received no further objections during the discussion period. We believe these unrefuted improvements justify a higher assessment, and we ask the Area Chair to evaluate the paper based on its current, strengthened state.

---


**Reviewer WTmm** commended the **novelty of decomposing the predictor space** into **interpretable** components, as well as our **theoretical guarantees** regarding consistent estimation, variance reduction, and closed-form fairness-accuracy trade-offs. The reviewer’s primary concerns regarding empirical validation appeared to stem from an inadvertent captioning error in Figure 3 and questions regarding experimental design.

In our revision, we clarified that our experimental design directly aligns with our main theoretical contribution: achieving the fairness-utility trade-offs established in Theorem 5.4. To further address the reviewer's request for validation of our theoretical claims, we **added experiments** confirming the asymptotic results in Proposition 5.2 and Lemma 5.3, as well as the variance reduction guaranteed by Theorem 5.1. Finally, we incorporated **new real-world data** (MEPS) to demonstrate practical effectiveness. We also emphasized that because our framework addresses the complex, under-explored setting of multivariate sensitive attributes, existing baselines are naturally limited; our work thus establishes a **rigorous benchmark for future research** in this domain. We believe these clarifications and additional experiments have completely addressed the reviewer's concerns, particularly given that no further objections were raised prior to the conclusion of the discussion period.


---

---

> ### Author Response · Authors · 2025-12-02
> **Author Summary of Revisions and Response to Reviewers 2**
>
> **Reviewer eWet** recognized the strength of our **theoretical contributions** and the **quality of our experiments**. However, the reviewer assigned a **borderline score (4)**, primarily citing the **linearity assumption** as a concern. We believe this assessment overlooks that linearity is a necessary condition for the strong theoretical guarantees we provide.
> In our response, we clarified that the **linear formulation is a deliberate design choice**, not an oversight. It allows us to **derive explicit, closed-form decompositions** and provable **fairness-utility trade-offs** in the challenging setting of **multivariate sensitive attributes** - results that are currently intractable in non-linear settings. We drew a direct analogy to the development of PCA: classical linear PCA had to be established as a rigorous baseline before extensions like Kernel PCA could be developed. Similarly, our work establishes the rigorous linear baseline for fair subspace decomposition. We have expanded Appendix C to outline how these principles extend to non-linear regimes (via kernels), but we maintain that the linear theory must be established first. Since all other comments were minor clarifications which we have fully addressed, and given the reviewer’s lack of further objection during the discussion period, we believe the strong theoretical contributions justify a positive decision.
>
> ---
>
> **Reviewer wL7E** strongly supported the paper (Rating: 8), highlighting the "**novel and principled decomposition**," "**strong theoretical results**" regarding asymptotic variance, and **"clear and compelling" empirical evidence**. The reviewer requested the discussion of the linear subspace assumption, along with a request for baseline implementation details. We addressed these points as follows: We expanded our **discussion to clarify that the linear assumption** is a deliberate methodological choice. It enables the derivation of closed-form, interpretable guarantees that are often required in high-stakes, regulated domains. We also added Appendix C to discuss how this linear foundation serves as a stepping stone for **non-linear extensions** (e.g., Kernel methods and neural networks). We addressed the question regarding baseline hyperparameters by highlighting that the official implementation for the key baseline (FRRM) has no hyperparameter to tune. We are grateful for the reviewer’s high assessment and excellent suggestions, which have helped us improve the clarity and reproducibility of the final manuscript.
>
> ---
>
> **Reviewer w8Fb** recognized the **novelty** of applying **envelope regression** to **fairness** and found the **synthetic experiments compelling**, noting a **"nice win in predictive MSE that one doesn't usually see in fairness papers."**
> The reviewer’s main questions concerned the framing of the method relative to causal and fair representation learning literature, as well as the interpretation of efficiency gains. We addressed these points as follows:
>
> - **Framing \& Related Work:** We clarified that our four-way subspace decomposition (predictive, sensitive, shared, and residual) is the key innovation driving our results. We added Appendix G to explicitly distinguish FERM from causal methods (which require strong assumptions like valid mediators that are unavailable in our setting) and fair representation learning (which is typically tailored for discrete attributes and lacks our closed-form optimality).
>
> - **Efficiency \& Interpretability:** We elaborated that our design choices are grounded in the **principle of statistical efficiency**: by explicitly removing residual noise irrelevant to the regression, FERM achieves the efficiency gains proven in Theorem 5.1. Regarding **interpretability**, we emphasized that FERM provides a refined decomposition of direct/indirect bias by explicitly identifying which subspaces contribute to prediction versus noise. We pointed out that this resolves a key limitation of prior approaches, which often apply statistically inefficient methods by treating all sensitive correlations uniformly.
>
> - **Presentation \& Background:** To address the request for more background, we added a new conceptual figure (Figure 4) to intuitively explain the envelope idea and improved the signposting to the technical details in the Appendix.
>
> - **Empirical Validation:** Crucially, we identified that the reviewer's hesitation regarding empirical performance was caused by the captioning error in Figure 3 (swapped subplots). With this correction, the efficiency gains are now unambiguous.
>
> We believe the initial rating of 4 does not resonate with the reviewer's own recognition of the paper's novelty and "compelling" synthetic results. Given that the primary empirical concern arose from a simple visualization error - which is now resolved - and that the reviewer did not raise any further objections during the discussion period, we believe the contributions of the paper justify a higher rating.

---

### Meta-Review · Area_Chair_oCFM · 2025-12-09

**Summary:**

This paper proposes a framework that leverages subspace decomposition via envelope regression to improve group fairness in regression algorithms. By decomposing the predictor space into four components and isolating the portion associated with sensitive attributes, the approach offers a principled way to distinguish between direct and indirect bias. Additionally, fairness in regression is an important topic to study and this paper takes an interesting step by using predictor-space decomposition. However, the reviewers pointed out several weaknesses of the paper, which I summarized in the next box.

**Reviewer Concerns:**

The authors actively engaged in the discussion period and submitted a revised paper. Some of the questions (e.g., framing, related work, etc) have been addressed. However, as several reviewers pointed out, the experiment section is a major limitation. The paper compares only against a single baseline (FRRM), and the reported performance improvements do not appear to be statistically significant. Additionally, the proposed method relies heavily on linearity assumptions which is a concern shared by most reviewers. Although I understand these assumptions are necessary for deriving theoretical guarantees (which are interesting results), I think it would be valuable to justify them through a more comprehensive empirical evaluation across a broader range of real-world datasets.

**Reviewer Scores:**

The two major concerns (baseline comparisons, linear assumptions) are not sufficiently addressed in my opinion. I suspect the reviewers might keep their scores if they had been able to participate during the discussion period

---

### Decision · Program_Chairs · 2026-01-26

Reject